

# α-pinene secondary organic aerosol at low temperature: Chemical composition and implications for particle viscosity

Wei Huang[1,2], Harald Saathoff[1], Aki Pajunoja[3], Xiaoli Shen[1,2], Karl-Heinz Naumann[1], Robert Wagner[1], Annele Virtanen[3], Thomas Leisner[1], Claudia Mohr[1,4,*]

[1]Institute of Meteorology and Climate Research, Karlsruhe Institute of Technology, Eggenstein-Leopoldshafen, 76344, Germany

[2]Institute of Geography and Geoecology, Karlsruhe Institute of Technology, Karlsruhe, 76131, Germany

[3]Department of Applied Physics, University of Eastern Finland, Kuopio, 80101, Finland

[4]Now at: Department of Environmental Science and Analytical Chemistry, Stockholm University, Stockholm, 11418, Sweden

*Correspondence to*: C. Mohr (claudia.mohr@aces.su.se)

**Abstract.** Chemical composition and viscosity of secondary organic aerosol (SOA) from α-pinene ($C_{10}H_{16}$) ozonolysis were investigated for low temperature conditions (223 K). Two types of experiments were performed using two simulation chambers at the Karlsruhe Institute of Technology, the Aerosol Preparation and Characterization chamber (APC), and the Aerosol Interaction and Dynamics in the Atmosphere chamber (AIDA). Experiment type 1 simulated SOA formation at upper tropospheric conditions: SOA was generated in the AIDA chamber directly at 223 K, 61 % relative humidity (RH) (experiment termed "cold humid", CH), or for comparison at 6 % RH (experiment termed "cold dry", CD) conditions. Experiment type 2 simulated SOA uplifting: SOA was formed in the APC chamber at room temperature (296 K), <1 % RH (experiment termed "warm dry", WD) or 21 % RH (experiment termed "warm humid", WH) conditions, and then partially transferred to the AIDA chamber kept at 223 K, and 61 % RH (WDtoCH) or 30 % RH (WHtoCH), respectively. Precursor concentrations varied between 0.7 and 2.2 ppm α-pinene, and 2.3 and 1.8 ppm ozone for type 1 and type 2 experiments, respectively. Among other instrumentation, a chemical ionization mass spectrometer (CIMS) with filter inlet for gases and aerosols (FIGAERO), deploying I⁻ as reagent ion, was used for SOA chemical composition analysis.

For type 1 experiments with lower α-pinene concentration and cold SOA formation temperature (223 K), smaller particles of 100−300 nm vacuum aerodynamic diameter ($d_{va}$) and higher mass fractions (>40 %) of adducts (molecules with more than 10 carbon atoms) of α-pinene oxidation products were observed. For type 2 experiments with higher α-pinene concentration and warm SOA formation temperature (296 K), larger particles (~500 nm $d_{va}$) with smaller mass fractions of adducts (<35 %) were produced.

We also observed differences (up to 20 ℃) in maximum desorption temperature ($T_{max}$) of individual compounds desorbing from the particles deposited on the FIGAERO Teflon filter for different experiments, indicating that $T_{max}$ is not purely a function of a compound's vapor pressure or volatility, but is also influenced by diffusion limitations within the particles (particle viscosity), interactions between particles deposited on the filter (particle matrix), and/or particle mass on the filter. Highest $T_{max}$ were observed for SOA under dry conditions and with higher adduct mass fraction; lowest $T_{max}$ for SOA under humid conditions and with lowest adduct mass fraction. The observations indicate that particle viscosity may be influenced by intra- and inter-molecular hydrogen bonding between oligomers, and particle water uptake, even under such low temperature conditions.

Our results suggest that particle physicochemical properties such as viscosity and oligomer content mutually influence each other, and that variation in $T_{max}$ of particle desorptions may provide implications for particle viscosity and particle matrix effects. The differences in particle physicochemical properties observed between our different experiments demonstrate the



importance of taking experimental conditions into consideration when interpreting data from laboratory studies or using them
as input in climate models.

## 1 Introduction

Atmospheric aerosols have adverse impacts on human health (Nel, 2005; Rückerl et al., 2011) and rank among the main drivers
of anthropogenic climate change (IPCC, 2013). Organic compounds make up a large fraction (20−90 %) of submicron
particulate mass (Zhang et al., 2007; Murphy et al., 2006; Jimenez et al., 2009; Ehn et al., 2014). Organic aerosol (OA) particles
can be directly emitted into the atmosphere from sources like fossil fuel combustion and forest fires (primary organic aerosol,
POA), or be formed in the atmosphere from the oxidation of gas-phase precursors (secondary organic aerosol, SOA). SOA
dominates the global budget of OA (Shrivastava et al., 2015). Gaseous SOA precursors (volatile organic compounds, VOC)
can be of both biogenic and anthropogenic origin. In the atmosphere, VOC are oxidized by the hydroxyl radical (OH), ozone
($O_3$), or the nitrate radical ($NO_3$) into semi-volatile, low volatile and/or extremely low volatile organic compounds (SVOC,
LVOC/ELVOC), which can partition into the particle phase and lead to the formation of SOA (Jimenez et al., 2009; Hallquist
et al., 2009; Jokinen et al., 2015; Ehn et al., 2014). Due to the wealth of precursors and formation mechanisms in both the gas
and particle phase, SOA is very complex and can contain thousands of compounds with a wide range of functionalities,
volatilities, and other physicochemical properties (Hallquist et al., 2009; Nozière et al., 2015).

Global estimates indicate that biogenic VOC emissions (539 Tg C $a^{-1}$) dominate over anthropogenic VOC emissions (16
Tg C $a^{-1}$), and that the global SOA production from biogenic VOC (22.9 Tg C $a^{-1}$) outreaches that from anthropogenic VOC
(1.4 Tg C $a^{-1}$) as well (Heald et al., 2008). An important class of biogenic VOC are monoterpenes ($C_{10}H_{16}$), emitted in
substantial amounts (43 Tg C $a^{-1}$, Heald et al., 2008) by vegetation (e.g. many coniferous trees, notably the pine). One of the
most abundant monoterpenes is α-pinene (24.8 % mass contribution to global monoterpene emissions, Kanakidou et al., 2005).
The fraction of total SOA mass from monoterpene oxidation products is estimated to be ~15 % globally, and can be higher in
some regions (e.g. in the boreal forest) (Heald et al., 2008).

SOA formation from α-pinene has been studied extensively in smog chambers (e.g. Kristensen et al., 2016; Denjean et al.,
2015; McVay et al., 2016), although studies covering a wide temperature range are rare (Saathoff et al., 2009; Donahue et al.,
2012). The reactions of α-pinene with $O_3$, OH- and $NO_3$- radicals lead to a large suite of oxygenated reaction products including
aldehydes, oxy-aldehydes, carboxylic acids, oxy-carboxylic acids, hydroxy-carboxylic acids, dicarboxylic acids, organic
nitrates etc. (Winterhalter et al., 2003; Kanakidou et al., 2005). Aerosol yields vary for the different oxidants, and the most
important process with regard to aerosol mass formation from the oxidation of α-pinene is the reaction with ozone (Kanakidou
et al., 2005).

The molecular formulae of organic species accounting for ~58−72 % of SOA mass from α-pinene ozonolysis have been
identified, and can largely be grouped into monomers ($C_{8-10}H_{12-16}O_{3-6}$, oxidation products from one α-pinene molecule) and
dimers ($C_{14-19}H_{24-28}O_{5-9}$, oxidation products from two α-pinene molecules) (Zhang et al., 2015). Autoxidation processes can
form highly oxidized molecules (HOM, elemental oxygen to carbon ratios of 0.7−1.3, Ehn et al., 2012), both monomers and
dimers, which have been shown to play an important role in atmospheric new particle formation (Ehn et al., 2014). Less
oxygenated dimers (e.g. esters and other accretion products), some of which have similarly low volatility as HOM and for
many of which formation mechanisms are still not known, are major products in aerosol particles from α-pinene ozonolysis,
and have been proposed as key components in organic particle growth in field and laboratory (Kristensen et al., 2014;
Kristensen et al., 2016; Tröstl et al., 2016; Zhang et al., 2015; Mohr et al., 2017).

SOA is a highly dynamic system between the gas and particle phase that continually evolves during residence time in the
atmosphere, becoming increasingly oxidized, less volatile, and more hygroscopic (Jimenez et al., 2009). As a consequence,
SOA residence time in the atmosphere at different temperature (T) and relative humidity (RH) conditions strongly influences





the particles' physicochemical properties such as phase state, and thus their effects on air quality and climate (Tsigaridis et al., 2006; Jimenez et al., 2009; Shiraiwa et al., 2017). Biogenic SOA has been shown to exist in phase states ranging from liquid to amorphous (semi-)solid in the atmosphere (Virtanen et al., 2010; Bateman et al., 2016; Shiraiwa et al., 2017). The phase state can affect gas uptake, gas-particle partitioning, diffusion, the particles' ability to act as cloud condensation nuclei (CCN) and/or ice nuclei (IN), as well as the particles' lifetime in the atmosphere (Shiraiwa et al., 2011; Price et al., 2015; Lienhard et

al., 2015). E.g. water diffusion coefficients in the water-soluble fraction of α-pinene SOA were measured for temperature conditions between 240 and 280 K. The results showed that water diffusion slowed down as temperature decreased, indicating increasing viscosity of SOA particles (Price et al., 2015). Diffusivity of organic molecules in SOA particles can show similar behavior, leading to large equilibration times under dry conditions (Shiraiwa et al., 2011) and/or cool conditions (Bastelberger et al., 2017). Observations of particle shape transformations (Järvinen et al., 2016), coalescence times (Pajunoja et al., 2014),

and the particle bounce factor (BF) (Virtanen et al., 2010; Pajunoja et al., 2015) are other parameters used to indicate the phase state and viscosity of particles. At dry conditions and at temperatures close to room temperature, the viscosity of α-pinene SOA is assumed to range from $10^5$ to (higher than) $10^8$ Pa s (Song et al., 2016; Renbaum-Wolff et al., 2013; Pajunoja et al., 2014), which corresponds to a semisolid state (Shiraiwa et al., 2011), whereas at a RH of about 90 % and room temperature its consistency is comparable to that of honey (~10 Pa s) (Renbaum-Wolff et al., 2013). Generally, SOA is more viscous in

cool and dry conditions (shown e.g. for α-pinene SOA at temperatures ranging from 235 K to 295 K and RH ranging from 35 to 90 %, Song et al., 2016; Järvinen et al., 2016; Shiraiwa et al., 2011; Wang et al., 2015; Kidd et al., 2014).

   Differences in α-pinene SOA chemical composition were observed for different SOA formation temperatures and RH conditions, such as lower oligomer content at higher RH (up to 87 %, Kidd et al., 2014), or lower temperature (285 K, Zhang et al., 2015). Given that the differences in physicochemical properties of SOA particles observed as a function of temperature

and RH only cover part of the range of atmospheric values, it is of great importance for our understanding of SOA climate effects to increase efforts in the investigation of SOA evolution at atmospherically relevant conditions, especially at low temperature. More knowledge on SOA at temperature and RH conditions that are representative of the upper troposphere, where SOA can be transported to or formed in-situ, is required in order to understand their potential importance for phase state, morphology, and chemical composition, and thus ultimately SOA cloud formation potential (Zhang et al., 2015; Virtanen

et al., 2010; Lienhard et al., 2015; Frege et al., 2017). However, such studies, particularly of SOA at low temperature, are still scarce.

   In the present work, we investigate the chemical composition and viscosity of α-pinene SOA formed at four different conditions corresponding to temperatures of 223 K and 296 K and RH between <1 % and 61 % in order to simulate SOA uplifting to and SOA formation in the upper troposphere. Samples for chemical ionization mass spectrometric analysis were

taken from the AIDA chamber at 223 K and collected on Teflon filters at two different times after starting the experiments. We discuss differences in these mass spectra and corresponding molecular desorption profiles when heating the filters from room temperature to 200 °C as well as possible implications for mutual interactions between particle chemical composition and viscosity.

## 2 Methodology

### 2.1 Environmental chambers and experimental design

The data for this study were acquired during a two-month measurement campaign (SOA15) in October and November 2015 at environmental chambers of the Institute of Meteorology and Climate Research (IMK) at the Karlsruhe Institute of Technology (KIT). The measurement campaign investigated yields, physical properties, and chemical composition of SOA from α-pinene ozonolysis as a function of precursor concentration, temperature and relative humidity (RH), as well as the ice




nucleation abilities of the SOA particles (Wagner et al., 2017). The focus on ice cloud formation allowed the investigation of the particles' physicochemical properties at temperatures as low as 223 K (representative of conditions in the upper troposphere at 8−12 km altitude in mid-latitudes), a range where detailed characterization is largely missing. Here, we discuss a subset of the SOA15 dataset that is based on experiments investigating the influence and mutual interaction of particle chemical composition and viscosity shortly after SOA formation and after a residence time of ~3.5 h which were formed at different temperature (223−296 K) and RH (<1−61 %) conditions using both environmental chambers available at IMK (see Fig. 1).

The AIDA (Aerosol Interaction and Dynamics in the Atmosphere) aerosol and cloud chamber is an 84.3 $m^3$ sized aluminum vessel. It can be operated in a temperature range of 183 to 333 K, a pressure range of 1 to 1000 hPa, RH from close to 0 to 200 %, and different warming and cooling rates (Schnaiter et al., 2016; Möhler et al., 2003; Saathoff et al., 2009).

The APC (Aerosol Preparation and Characterization) chamber (Möhler et al., 2008) is a 3.7 $m^3$ sized stainless steel vessel, right next to AIDA and connected with it. The APC chamber can only be operated at room temperature (296 K) and was used to prepare SOA particles in a reproducible manner.

We present two types of chamber experiments (Fig. 1): For the first type, SOA from α-pinene ozonolysis was directly formed at 223 K in the AIDA chamber. For the second type, SOA was first produced in the APC chamber kept at room temperature and then transferred to the AIDA chamber kept at 223 K. The second type of experiment thus represents a simplified simulation of particle formation in the boundary layer and subsequent uplifting of particles to higher altitudes with lower temperature conditions. We stress here that for both types of experiments, the particles were sampled from the cold AIDA chamber for chemical analysis. The detailed conditions for these two types of experiments are listed in Table 1. During the first type of chamber experiment, SOA was formed by reaction of an excess of ozone (initially 2.3 ppm generated by silent discharge in pure oxygen, Semozon 030.2 discharge generator, Sorbios GmbH) with α-pinene (initially 0.714 ppm, 99 %, Aldrich) in the dark AIDA chamber at 223 K, 61 % RH (experiment termed "cold humid", CH) or 6 % RH (experiment termed "cold dry", CD) conditions. For the second type of chamber experiment, SOA was first formed with an excess of ozone (initially 1.8 ppm) and 2.2 ppm α-pinene in the dark APC chamber at room temperature (296 K), <1 % RH (experiment termed "warm dry", WD) or 21 % RH (experiment termed "warm humid", WH) conditions. After a residence time of 1−1.5 h in the APC chamber, a fraction of the SOA particles was then transferred to the dark AIDA chamber kept at 223 K, 61 % RH (WDtoCH) or 30 % RH (WHtoCH), respectively, resulting in the particle number concentrations ranging between 1500−2200 $cm^{-3}$ in the AIDA chamber. No OH scavenger was used during SOA formation, and RH was kept constant in AIDA during the course of the experiments. The time series of total particle mass for experiment type 1 (particles formed in-situ in AIDA, CH) and experiment type 2 (aerosols formed in APC and transferred to AIDA, WDtoCH) are shown in Fig. 2. The times t0 (right after SOA formation (CD, CH) or SOA transfer (WDtoCH, WHtoCH)) and t1 (~3.5 h later) indicate the points in experiment time which were used for the investigation of the physicochemical evolution of α-pinene SOA.

## 2.2 Temperature and relative humidity measurements

Temperature (T) in the AIDA chamber was measured by home-made thermocouples (NiCrNi) and home-made PT 100 temperature sensors, which are regularly calibrated with reference sensors traceable to standards of the National Institute of Standards and Technology (NIST). Under static conditions, gas temperature in the AIDA chamber deviated by less than 0.3 K in time and in space. Water vapor concentrations in the AIDA chamber were measured by a home-made tunable diode laser (TDL) spectrometer with an accuracy of $\pm 5$ % (Fahey et al., 2014; Skrotzki et al., 2013), as well as a dew point mirror hygrometer (MBW373LX, MBW Calibration Ltd.) traceable to different national metrology standards including Federal Institution of Physical Technology (PTB), National Physical Laboratory (NPL), Federal Office of Metrology and Surveying (BEV) and NIST. Relative humidity (RH) in the AIDA chamber was calculated using the measured water vapor concentrations and temperature based on the saturation water vapor pressures given by Murphy and Koop (Murphy and Koop, 2005).



### 2.3 Particle and gas measurements

Number concentrations of SOA particles formed in APC or AIDA were recorded with two condensation particle counters (CPC3022, CPC3010, TSI Inc.) outside the temperature controlled housing of the chambers via stainless steel tubes extending 35 cm into the AIDA chamber. The absolute uncertainty of the number concentrations is estimated to be $\pm 20\%$ by comparison of the different CPCs with each other and with an electrometer (3068, TSI Inc.). Particle size distributions were sampled in the same way from both chambers with scanning mobility particle sizers (SMPS, DMA 3071 connected to a CPC 3010, TSI Inc.). Mass concentrations were derived from integrated number size distributions and their conversions to mass using their corresponding calculated particle density (1.3−1.5 kg m$^{-3}$). Particle densities were calculated using the ratio of vacuum dynamic diameter ($d_{va}$) measured by a high-resolution time-of-flight aerosol mass spectrometer (HR-ToF-AMS, Aerodyne Research Inc., hereafter AMS) and mobility diameter ($d_m$) measured by the SMPS, assuming particle sphericity (shape factor = 1). Ozone concentrations were measured by an ozone monitor (O$_3$ 41M, Environment S.A.). The AMS was connected to the AIDA chamber by a stainless steel tube of 1.35 m length (flowrate 0.1 l min$^{-1}$, residence time 1.6 s). It was equipped with a high-pressure lens (HPL) (Williams et al., 2013) and continuously measured total organic particle mass as a function of size (up to 2.5 μm particle aerodynamic diameter) at a time resolution of 0.5 min. Elemental oxygen to carbon (O:C) and hydrogen to carbon (H:C) ratios were derived using the EALight_1_06 procedure in the AMS data analysis software package SQUIRREL (version 1.57H) (Canagaratna et al., 2015). An AMS collection efficiency (CE) of 0.4−0.5 was used, except for the CH experiment where CE was 0.7, likely due to higher particle water content. AMS mass concentrations compare well with the total mass derived from SMPS. Individual organic compounds in both the gas and particle phase were measured with a Filter Inlet for Gases and AEROsols coupled to a high-resolution time-of-flight chemical ionization mass spectrometer (FIGAERO-HR-ToF-CIMS, Aerodyne Research Inc., hereafter CIMS) deploying iodide ions (I$^-$) as reagent ions (Lopez-Hilfiker et al., 2014; Lee et al., 2014). With the FIGAERO, gases were measured via a 0.83 m Fluorinated Ethylene Propylene (FEP) tube while particles were simultaneously collected on a Teflon filter via a separate sampling port (stainless steel tube of 0.66 m length, flowrate 5 l min$^{-1}$, residence time 0.9 s). At regular intervals (5−20 min, see Table 2), the gas phase measurement was switched off and particles on the filter were desorbed by a flow of ultra-high purity (99.999 %) nitrogen heated from room temperature to 200 °C over the course of 35 min. The resulting mass spectral signal evolutions as a function of desorption temperature are termed thermograms (Lopez-Hilfiker et al., 2014). Single mode thermograms of a compound with signal maxima occurring at distinct desorption temperatures ($T_{max}$), which correlate with the compound's enthalpy of sublimation, can be used to infer its saturation vapor pressure (Lopez-Hilfiker et al., 2015; Mohr et al., 2017). Multi-mode thermograms indicate contributions from isomers having different vapor pressures, or thermal fragmentation of larger molecules during the heating of the filter (Lopez-Hilfiker et al., 2015). Integration of thermograms of individual compounds yielded their total signal in counts per deposition, which were converted to mass concentrations using a sensitivity of 22 counts s$^{-1}$ ppt$^{-1}$ (collisional limit, Lopez-Hilfiker et al., 2016). For each experiment, backgrounds were determined by sampling from the AIDA chamber before adding any precursor gases.

All instruments were set up at room temperature, outside the temperature-controlled housing of AIDA. Despite inlet insulation with Armaflex, we calculated a theoretical temperature increase of ~15 K for the particle inlet of the CIMS (the FIGAERO filter was thus presumably at 238 K during deposition), and cannot entirely rule out partial evaporation of water or semivolatile organic compounds, which is taken into account in our interpretation of results.

### 3 Results and discussion

### 3.1 Organic particle mass and size distribution



Figure 2A−B shows the time series of total particle mass derived from SMPS size distributions, total organic particle mass
       measured by AMS, and total mass of particulate oxygenated hydrocarbons ($C_{x>1}H_{y>1}O_{z>1}$ detected as clustered with I⁻, termed
       CHOI compounds) measured by CIMS for both types of experiments. Panel A depicts the CH experiment, representative of
       experiment type 1, where particles were directly formed in AIDA. Panel B represents experiment type 2, where aerosol was
       formed in the APC and transferred to AIDA (here the WDtoCH example, see Table 1). Please note that the data were not wall-
loss corrected. Gaps in the AMS time series were due to filter measurements. To investigate the evolution of the SOA particles'
       physicochemical properties with time, we chose two points in time during the experiments, t0 and t1. t0 is the first FIGAERO
       filter deposition from AIDA after particle formation (experiment type 1) or particle transfer (experiment type 2), while t1 is
       approximately 3.5 h later. Averaged concentrations of total organics, total CHOI compounds, as well as elemental oxygen to
       carbon (O:C) ratios at t0 and t1, together with an overview of the experimental conditions including temperature (T), relative
humidity (RH), and added precursor (α-pinene and O₃) concentrations for all experiments discussed here (WDtoCH, WHtoCH,
       CH, and CD) are listed in Table 1. Particle size distributions measured by AMS for all four experiments at t0 and t1 are shown
       in Fig. 2C-D.

       For SOA formed in AIDA (type 1 experiments), mean total organic mass concentrations at t0 and t1 were in the range of
       67.5−440.1 µg m⁻³, and mean total concentrations of CHOI compounds 97.8−247.6 µg m⁻³. When particles were transferred
from the APC chamber (type 2 experiments), organic and CHOI mass concentrations in AIDA reached values of 48.5−64.2
       µg m⁻³ and 23.3−40.7 µg m⁻³, respectively. We stress here that even though particle mass concentrations in AIDA were higher
       for the experiments of type 1 (particles formed at 223 K directly in AIDA), the α-pinene concentration for the type 2
       experiments was higher by a factor of ∼3 (Fig. 2A−B and Table 1). This also led to larger particle sizes for the type 2
       experiments. Due to additional α-pinene addition between t0 and t1 only for the CH experiment, we observed a step increase
of total particle mass for this experiment (Fig. 2A).

       The discrepancies between AMS and CIMS concentrations are likely due to the CIMS with I⁻ as reagent ion being more
       sensitive to more polar oxygenated organic compounds (Lee et al., 2014) and thus only a potential subset of organic compounds
       measured by AMS. Evaporation losses of particulate compounds during filter deposition in the FIGAERO may play a minor
       role. In addition, by using the collisional limit for the CIMS data, we apply maximum sensitivity and thus present lower limits
of CHOI compounds. The differences between the AMS and SMPS derived mass concentrations in Fig. 2A are likely due to
       the lower transmission of sub-200-nm particles in the aerodynamic lens of the AMS used here. The AMS measured lower
       concentrations than the SMPS at the beginning of the CH experiment (Fig. 2A), when the newly formed particles were much
       smaller (see Fig. 2C) compared to later in the experiment when they had grown in size (see Fig. 2D). For the WDtoCH
       experiment (Fig. 2B) with larger particles transferred from the APC to the AIDA chamber, AMS and SMPS derived mass
concentrations agree very well. The slightly decreasing trend observed during both experiments was due to wall losses
       (Donahue et al., 2012).

## 3.2 Chemical characterization of SOA particles

### 3.2.1 Elemental oxygen to carbon ratios

       Elemental oxygen to carbon (O:C) ratios were calculated using both AMS and CIMS data. The mean AMS O:C ratios for SOA
formed in APC and AIDA were 0.34−0.36 and 0.26−0.30, respectively (Table 1). This is representative of O:C ratios for
       relatively fresh SOA measured in ambient studies (Mohr et al., 2012; Ge et al., 2012; Canagaratna et al., 2015). For CHOI
       compounds measured by CIMS, the calculated mean O:C ratios for SOA formed in APC and AIDA were 0.59−0.66 and
       0.56−0.61, respectively. The AMS O:C ratio is expected to be lower than that of the CHOI compounds measured by iodide
       CIMS, as the latter is selective towards polar oxygenated compounds. The potential loss of semivolatiles from the filter during
FIGAERO deposition may additionally increase the mass-averaged O:C ratio of compounds measured with this instrument.





The O:C ratios of SOA formed in the APC were slightly higher than those formed in AIDA, likely a result of the difference in precursor concentrations and temperature and thus partitioning behavior of semivolatile SOA compounds during formation between the particles and chamber walls. We rule out a dilution effect when transferring particles from APC to AIDA, since the dilution factor was orders of magnitude smaller than the decrease in saturation vapor pressure due to the temperature

reduction from APC (296 K) to AIDA (223 K), and confirmed by the absence of a change in particle size after transfer. For all experiments, O:C ratios remained largely constant from t0 to t1.

### 3.2.2 FIGAERO-CIMS mass spectra

Mass spectra of integrated desorptions from the CIMS are compared for the four experiments and two points in time, t0 and t1. Mass spectra shown were normalized to the sum of signal of all detected CHOI compounds. The corresponding mass

loadings and sampling times (particle collection on filter) for the four experiments are listed in Table S1. Figure 3A shows a comparison of mass spectral patterns for the experiments WDtoCH and CD, 3B for WHtoCH and CD, and 3C for CH and CD, all at t0 (the same comparisons for t1 are to be found in Fig. S1). Overall, the mass spectral patterns across all experimental conditions and points in time were relatively similar. Monomers ($C_mH_yO_z$ compounds, $m \leq 10$), dimers ($C_nH_yO_z$ compounds, $11 \leq n \leq 20$), and even trimers ($C_pH_yO_z$ compounds, $21 \leq p \leq 30$) clustered with $I^-$ were observed in the mass spectra at t0 and t1

for all occasions.

Monomers dominated the overall signal of detected compounds, with the biggest signal at m/z 327 (mainly $C_{10}H_{16}O_4I_1^-$, likely hydroxy-pinonic acid clustered with $I^-$). As we can see from Fig. 3, relatively higher contributions of monomers were measured at t0 for experiments WDtoCH and WHtoCH compared to CD. The difference in relative monomer contributions for experiments CH and CD was less distinct. At the same time, relatively larger contributions from dimers and trimers (inserts

in Fig. 3) were observed for the experiment CD (and to a lesser extent for the CH). This was also the case for t1 (Fig. S1).

Figure 4 shows the relative mass contributions of monomers and adducts (including dimers and trimers) for the four experiments at both time points. As already observed in the mass spectral patterns, larger relative mass contributions from monomers were measured for the type 2 experiments (WDtoCH, WHtoCH), and larger relative mass contributions from adducts for the type 1 experiments (CH, CD). There was no significant change for the relative contributions and absolute

concentrations of adducts (Fig. S2) between t0 and t1 for type 2 experiments (WDtoCH, WHtoCH). For type 1 experiments (CH and CD), absolute concentrations of monomers and adducts (Fig. S2) were increasing from t0 to t1 due to the addition of α-pinene after t0 and hence the continuing production of oxidation products and particle mass (compare Fig. 2). However, the relative contributions of monomers for type 1 experiments were increasing from t0 to t1, which may be partially influenced by smaller FIGAERO sampling time and thus less evaporation losses of semivolatiles at t1 (see Table 2 and supplement), but

mostly by increased condensation of semivolatiles or lower-molecular-weight products with increasing particle size (compare Fig. 2C−D).

Figure 5 shows the average mass-weighted number of carbon atoms (numC) and oxygen atoms (numO) for CHOI compounds for the four experiments at t0 and t1. The corresponding average mass-weighted compounds' formulae for SOA generated in APC and AIDA were $C_{10-12}H_yO_{6-7}$ and $C_{11-13}H_yO_{6-7}$, respectively. Slightly bigger numC were observed for type

1 experiments (CH, CD) than type 2 experiments, with the largest value for experiment CD, followed by CH and WHtoCH. NumC was smallest for WDtoCH. There was no obvious trend for numO.

In summary, smaller particles with slightly lower O:C ratios, bigger carbon numbers and relatively more mass from adducts were observed for type 1 experiments (CH, CD), which had lower α-pinene concentration and colder formation temperature (223 K) compared to the type 2 experiments. For type 2 experiments (WDtoCH, WHtoCH), higher α-pinene concentration (by

a factor of ~3) and warmer formation temperature (296 K) produced larger particles with slightly higher O:C ratios, smaller carbon numbers and relatively more mass from monomers. The slightly higher O:C ratio in type 2 experiments is thus not due to bigger oxygen numbers, but due to smaller carbon numbers (Fig. 5), indicating that relatively more small oxygenated





molecules were formed for type 2 experiments. This is likely due to higher α-pinene concentration and faster oxidation at 296 K leading to rapid condensation of monomers, providing enough gaseous oxidation products for the equilibrium of

semivolatiles to be shifted to the particle phase. Type 1 experiments, on the other hand, were performed with lower α-pinene concentration, and particles were formed in-situ, favoring higher contributions of larger ELVOC/LVOC compounds, especially at the early stages of particle growth (Tröstl et al., 2016). At the same time, the low temperature conditions may also have shifted equilibrium to the particle phase and led to condensation of compounds with relatively lower degree of oxygenation (compared to warm temperature conditions). Overall, the differences observed in mass spectral patterns between

the two types of experiments are a consequence of both temperature and precursor concentration differences. They underline the importance of experiment conditions when interpreting laboratory data or modelling.

### 3.3 Thermograms: Variation in $T_{max}$ of SOA compounds for different experiments

In addition to information on mass spectral patterns and mass loadings when peaks are integrated, the FIGAERO also provides signal curves as a function of desorption temperature (referred to as thermograms). These thermograms result from the thermal desorption of particles deposited on the Teflon filter in the FIGAERO (Lopez-Hilfiker et al., 2014). Single mode thermograms of a compound with a signal maximum occurring at a distinct desorption temperature ($T_{max}$) can be used to infer the compound's saturation vapor pressure (Lopez-Hilfiker et al., 2015; Mohr et al., 2017). However, evaporative behavior and inferred volatility of a particle-bound compound are also influenced by the particles' physical phase state, particle phase diffusivity and viscosity (Yli-Juuti et al., 2017). Here we show that thermograms may also be used for qualitative information

on particle viscosity.

Thermograms resulting from the thermal desorption of deposited SOA particles from the four experiments CH, CD, WDtoCH, and WHtoCH at both time points t0 and t1 were analyzed. Examples of the thermograms of a monomer ($C_{10}H_{16}O_4$, molecular formula corresponding to hydroxy-pinonic acid), and an adduct ($C_{17}H_{26}O_8$, molecular formula identified in SOA from α-pinene ozonolysis by e.g. Zhang et al., 2015; Mohr et al., 2017) both clustered with I$^-$ at t0 are shown in Fig. 6A−B.

Panel C shows the sum of thermograms of all monomers, panel D the sum of all adduct thermograms at t0. The same plots for t1 can be found in Fig. S3. Thermograms and sums of thermograms were normalized to their maximum values. The corresponding mass loadings and sampling times (particle collection on filter) for the four experiments are listed in Table 2. For experiment CD, the $C_{10}H_{16}O_4I_1^-$ thermograms exhibited a multi-modal shape, indicative of contributions from isomers having different vapor pressures, or thermal decomposition of larger molecules. Based on previous FIGAERO data analyses

(Lopez-Hilfiker et al., 2015; D'Ambro et al., 2017; Wang et al., 2016), we can safely presume that the first mode corresponds to the monomer.

Figure 6A−B shows that $T_{max}$ of an individual compound varied by up to 20 °C, depending on experimental conditions. It has been shown earlier that thermograms and corresponding $T_{max}$ are highly reproducible for stable conditions (Lopez-Hilfiker et al., 2014). In our instrument, $T_{max}$ varied by 2 °C at most for the monomer, $C_{10}H_{16}O_4$, and another adduct, $C_{16}H_{24}O_6$

(molecular formula identified in SOA from α-pinene ozonolysis by e.g. Zhang et al., 2015) both clustered with I$^-$, for 6 subsequent thermograms under stable conditions, respectively (Fig. S4). The variation in $T_{max}$ as a function of experiment types observed here thus indicates that the shape of thermogram for a given compound and given FIGAERO configuration is not only defined by the compound's enthalpy of evaporation. For both $C_{10}H_{16}O_4I_1^-$ and $C_{17}H_{26}O_8I_1^-$ thermograms, $T_{max}$ was highest for experiment CD, followed by WHtoCH, and CH, WDtoCH. Similar trends were observed for all compounds

measured by the FIGAERO-CIMS, as shown by the sums of thermograms of all monomer compounds (Fig. 6C), or sums of thermograms of all adduct compounds (Fig. 6D). Sum $T_{max}$ of monomers and adducts varied from 46 °C (experiment WDtoCH) to 74 °C (experiment CH) and 93 °C (experiment WHtoCH) to 104 °C (experiment CD), respectively.

Variation in $T_{max}$ of the sum of CHOI compounds was larger for monomers (Fig. 6C) than for adducts (Fig. 6D). Monomers are thus the more important contributors to the shifts in $T_{max}$, likely because at the higher temperatures where adducts desorb,




particle matrix effects may become less important. Since the sum of thermograms and its $T_{max}$ is highly influenced by compounds with large signal, we also show a box and whisker diagram of $T_{max}$ for monomers and adducts (Fig. S5). The median $T_{max}$ values showed similar variation as the $T_{max}$ values based on thermogram sums. Examples of the $T_{max}$ distribution of individual CHOI compounds in numO/numC space at t0 are shown in Fig. 7 for the WDtoCH and CD experiments. Points were color-coded by $T_{max}$. Compounds with nominal molecular formulae $C_{8-10}H_yO_{4-6}I^-$ were the main contributors to mass

concentrations (data not shown), and thus also aggregated $T_{max}$ values. Generally, $T_{max}$ for CHOI compounds ranged from 25 to 165 °C, and increased with carbon numbers and oxygen numbers of compounds, as is to be expected given the relationship between enthalpy of evaporation and volatility of a compound (Lopez-Hilfiker et al., 2015; Mohr et al., 2017). The comparison between WDtoCH (Fig. 7A) and CD (Fig. 7B) experiments, however, showed differences in $T_{max}$ values for most compounds. $T_{max}$ values, especially of a lot of compounds with nominal molecular formulae $C_{5-10}H_yO_{1-10}I^-$ and $C_{15-25}H_yO_{11-20}I^-$, were

higher for the CD experiment. The similar behavior in variation in $T_{max}$ of most compounds measured by FIGAERO-CIMS indicates that $T_{max}$ is not purely a function of the compounds' vapor pressures or volatilities, but is influenced by diffusion limitations within the particles (particle viscosity) (Vaden et al., 2011; Yli-Juuti et al., 2017), interactions between particles deposited on the filter (particle matrix), and/or particle mass on the filter. In the following we will discuss these implications in more detail.

Mass transport limitations within SOA particles, often measured or modelled as evaporation rates of specific compounds (Yli-Juuti et al., 2017; Wilson et al., 2015; Roldin et al., 2014), have been related to the particle viscosity (Vaden et al., 2011; Yli-Juuti et al., 2017). Particle viscosity is highly influenced by temperature and relative humidity (RH) (Shiraiwa et al., 2017; Kidd et al., 2014), with higher viscosities at cool and/or dry conditions (Shiraiwa et al., 2011). Since the temperature was 223 K in AIDA for all experiments discussed here, the observed differences in $T_{max}$, and presumed viscosity, cannot be directly

explained by differences in temperature. In addition, during desorption of compounds with the FIGAERO, particles are actively heated (with heat transfer assumed to be immediate), and are not evaporating under equilibrium conditions. Presumed variations in particle viscosity based on observed variations in $T_{max}$ must therefore be due to variations in particle chemical composition, and/or RH differences.

    The biggest $T_{max}$ difference in Fig. 6 was between WDtoCH and CD experiments, which was in accordance with the largest

differences in mass spectra as discussed above (see Fig. 3A and 4). This is indicative of a relationship between $T_{max}$ in the thermograms and particle chemical composition. It has been shown earlier that the chemical properties of particulate compounds influence particle viscosity (Kidd et al., 2014; Hosny et al., 2016). Viscosity is expected to be higher with higher oligomer content, due to inter-component hydrogen bonding, especially at low RH (Kidd et al., 2014). This is in accordance with our results, which showed highest $T_{max}$ values for the CD experiment, which also had the highest contribution from

adducts.

    RH is an additional parameter that greatly influences particle viscosity (Kidd et al., 2014; Hosny et al., 2016; Renbaum-Wolff et al., 2013). Even though the fact that the SOA particles might be dried very quickly by the dry heated nitrogen during particle desorption, we suppose that RH might have a "memory effect" and still influence $T_{max}$. RH conditions during the four experiments presented here ranged from 6 % (CD), 30 % (WHtoCH) to 61 % (WDtoCH and CH). Note that these were the

conditions of the measurement time in the AIDA chamber; for WDtoCH and WHtoCH, the RH conditions during SOA formation in the APC chamber were 1% and 21%, respectively. We thus need to differentiate between $RH_{formation}$ and $RH_{measurement}$. As shown in Fig. 8, there was no trend between $RH_{formation}$ and $T_{max}$, indicating that the RH during particle formation did not play an important role in the observed viscosity variation. However, we observed a negative correlation of $RH_{measurement}$ and $T_{max}$ of all monomer compounds at t0, indicating that even under low temperature conditions of 223 K there

is particle water uptake, and an influence of RH on viscosity. Calculated particle water content derived from AMS measurements is prone to large uncertainties (Engelhart et al., 2011); we observed a qualitative positive correlation with RH (data not shown). Particle water uptake thus seems to influence particle viscosity even at such low temperature and on such





short timescales (few hours). To what extent RH and particle water uptake, or chemical properties and adduct content, and their respective influence on water uptake via increased hygroscopicity, contribute to the observed differences in $T_{max}$ and presumed viscosity, we can only speculate. In the CH and WDtoCH experiments, $RH_{measurement}$ was ~60 % for both. The adduct mass fraction was only slightly higher for SOA in the CH experiment, and so was $T_{max}$ and thus potentially particle viscosity. More controlled studies at low temperature are needed to separate these effects.

We also noticed that different mass loadings on the filter due to different sampling times and/or sample concentrations influenced the shape of thermograms and thus $T_{max}$. $T_{max}$ increased as a function of mass loading on the filter, likely due to the increase in heat capacity of the increasing mass of the particle matrix, and potential interactions between the particles. The dependency of $T_{max}$ on filter mass loading was not linear, and for our FIGAERO reached a plateau at mass loadings of 2−4 µg. Our results are therefore not affected by the mass loading effect, but we recommend taking it into account in analyses that involve $T_{max}$. A detailed discussion can be found in the supplement.

## 4 Conclusions and atmospheric implications

In this study, α-pinene SOA physicochemical properties such as chemical composition, phase state, and viscosity were investigated at low temperature (223 K) and different relative humidity (RH) using two simulation chambers (APC and AIDA). Two types of experiments were performed: For type 1 experiments, SOA was directly generated in the AIDA chamber kept at 223 K, 61 % RH (experiment termed "cold humid", CH) or 6 % RH (experiment termed "cold dry", CD) conditions. For type 2 experiments, SOA was formed in the APC chamber at room temperature (296 K), <1 % RH (experiment termed "warm dry", WD) or 21 % RH (experiment termed "warm humid", WH) conditions, and then partially transferred to the AIDA chamber kept at 223 K, 61 % RH (WDtoCH) or 30 % RH (WHtoCH) conditions, respectively, to simulate SOA uplifting.

For type 1 experiments (CH, CD) with lower α-pinene concentration and cold SOA formation temperature (223 K), smaller particles with relatively more mass from adducts were observed. For type 2 experiments (WDtoCH, WHtoCH) with higher α-pinene concentration (by a factor of ~3) and warm SOA formation temperature (296 K), larger particles with relatively more mass from monomers were produced. The differences observed in mass spectral patterns between the two types of experiments are likely a consequence of both temperature and precursor concentration differences. Higher α-pinene concentration and faster oxidation at 296 K during SOA formation in the APC chamber shift the gas-particle equilibrium to the particles, resulting in larger mass fractions of semivolatile and/or monomer compounds. Low temperature conditions in the AIDA chamber during SOA formation on the other hand may result in condensation of compounds with a relatively lower degree of oxygenation. Our results show that depending on where SOA formation takes place in the atmosphere (e.g. boundary layer or upper troposphere), chemical properties can vary, and with that reactivity and lifetime.

In addition to the differences in mass spectral patterns for the different experiments, we also observed differences in the shape of thermograms resulting from the desorption of SOA particles collected on the FIGAERO filter: $T_{max}$ of an individual compound in the thermograms varied by up to 20 °C depending on experimental conditions, indicating that $T_{max}$ is not only influenced by a compound´s vapor pressure or volatility, but also by diffusion limitations within the particles (particle viscosity). For both $C_{10}H_{16}O_4I^-$ and $C_{17}H_{26}O_8I^-$ thermograms, $T_{max}$ was highest for experiment CD, followed by WHtoCH, and CH, WDtoCH. We observed higher $T_{max}$ for α-pinene SOA particles with higher oligomer mass fractions, indicating the potential role of intra- and inter-molecular hydrogen bonds between these large and highly functionalized molecules for the increase in particle viscosity (Kidd et al., 2014). Furthermore, $T_{max}$ was negatively correlated with RH in the particle reservoir and particle water content, suggesting that hygroscopic properties and water uptake are important factors even at such low temperature. We also demonstrated an effect of mass deposited on the FIGAERO filter on $T_{max}$, which needs to be taken into account for further studies relying on $T_{max}$.




The results suggest that particle physicochemical properties such as viscosity and oligomer content mutually influence each other. More controlled experiments at low temperature are needed to separate the direct effects of RH and particle water

uptake, or chemical properties such as adduct content, and the indirect effects of chemical properties on water uptake via changes in hygroscopicity on the observed differences in $T_{max}$ and presumed viscosity. The differences in SOA physicochemical properties observed in our set of experiments as a function of temperature, RH, and precursor conditions demonstrates the importance of ambient and laboratory measurements at a wide range of atmospherically relevant conditions, and of taking experimental conditions into careful consideration when interpreting laboratory studies or using them as input in

climate models.

**Data availability**

Data are available upon request to the corresponding author.

**Author contributions**

W.H., H.S., A.P., X.S., K.-H.N., A.V., T.L., and C.M. designed research; W.H., H.S., A.P., X.S., R.W., and C.M. performed
research; W.H., H.S., A.P., X.S., and C.M. analyzed data; and W.H. and C.M. wrote the paper.
The authors declare no conflict of interest.

**Acknowledgements**

Technical support by the AIDA staff at IMK-AAF, and financial support by European Research Council (ERC-StG QAPPA 335478), Academy of Finland (259005 and 272041) and by the China Scholarship Council (CSC) for W.H. and X.S., are
gratefully acknowledged.

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





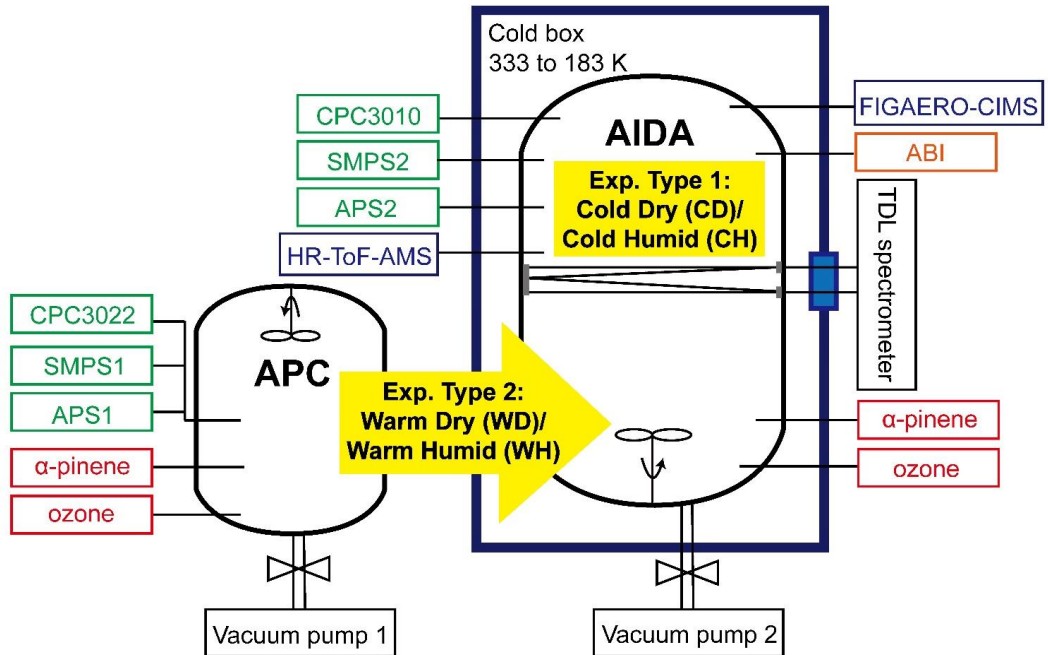

**Figure 1.** Brief schematic and conditions for the two types of experiments (modified from Wagner et al., 2017). Both chambers at IMK (APC and AIDA) were used in this study. Instruments are annotated in green, blue or orange, and precursor gases in red. More detailed information for the instruments and precursor gases are explained in the text.

640



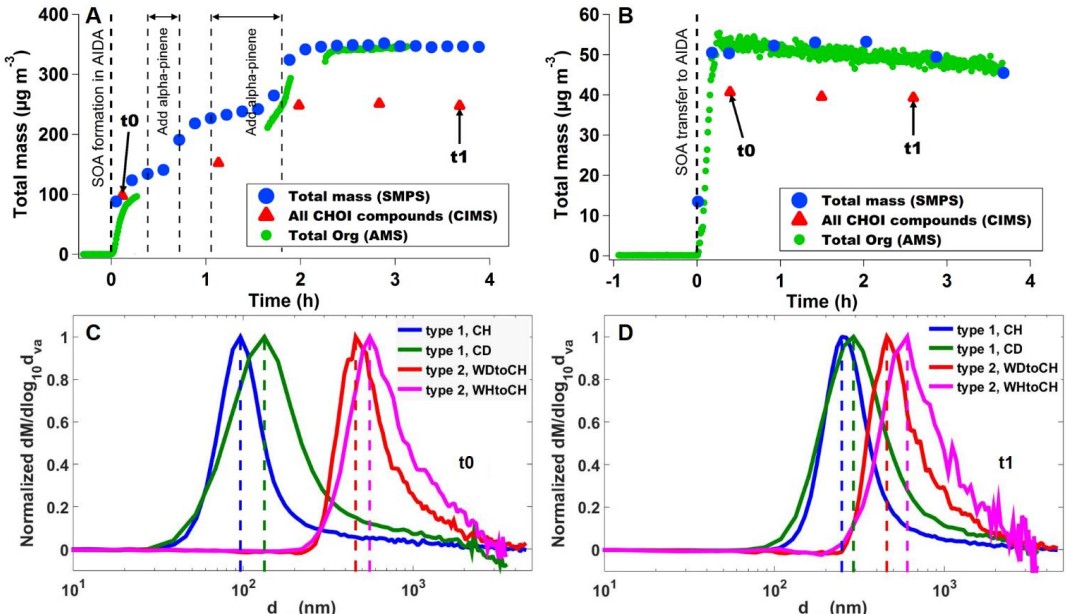

**Figure 2.** (A) Particle mass concentrations derived from SMPS size distributions (blue circles), CHOI mass concentrations measured by CIMS (red triangles), and organic mass concentrations measured by AMS (green circles) representative for type 1 experiments (here CH), (B) representative for type 2 experiments (here WDtoCH). Data were not wall loss corrected. t0 and t1 indicate points in time used for comparisons in this study. (C) Averaged size distributions measured by AMS at t0, and (D) t1 for the four experiments.





**Table 1.** Experimental conditions and precursor concentrations for the four experiments discussed in this study: CH, CD (type 1), and WDtoCH, WHtoCH (type 2). Total organic mass, CHOI mass concentrations, and elemental oxygen to carbon (O:C) ratios are given for t0 and t1. Relative humidity (RH) values from the APC chamber were measured at room temperature (296 K).

| Exp. name | SOA position | T (K) | RH (%) | $\alpha$-pinene added (ppm) | Ozone added (ppm) | Total Org ($\mu$g m$^{-3}$) | Total CHOI ($\mu$g m$^{-3}$) | O:C |
|---|---|---|---|---|---|---|---|---|
| CH | AIDA | 223 | 61.0 | 0.714 | 2.3 | 67.5/319.5 | 97.8/247.6 | 0.26/0.30 |
| CD | AIDA | 223 | 6.02 | 0.714 | 2.3 | 260.1/440.1 | 110.6/160.4 | 0.28/0.29 |
| WDtoCH | APC $\rightarrow$ AIDA | 296 $\rightarrow$ 223 | <1 $\rightarrow$ 60.6 | 2.2 | 1.8 | 50.9/48.5 | 40.7/39.3 | 0.34/0.34 |
| WHtoCH | APC $\rightarrow$ AIDA | 296 $\rightarrow$ 223 | 21 $\rightarrow$ 30.3 | 2.2 | 1.8 | 64.2/58.4 | 23.4/23.3 | 0.36/0.37 |





650

**Figure 3.** FIGAERO-CIMS mass spectra (normalized to the sum of signal of all detected CHOI compounds) of experiments WDtoCH and CD (A), WHtoCH and CD (B), CH and CD (C) at t0. Inserts show enlarged regions of dimers (left) and trimers (right).





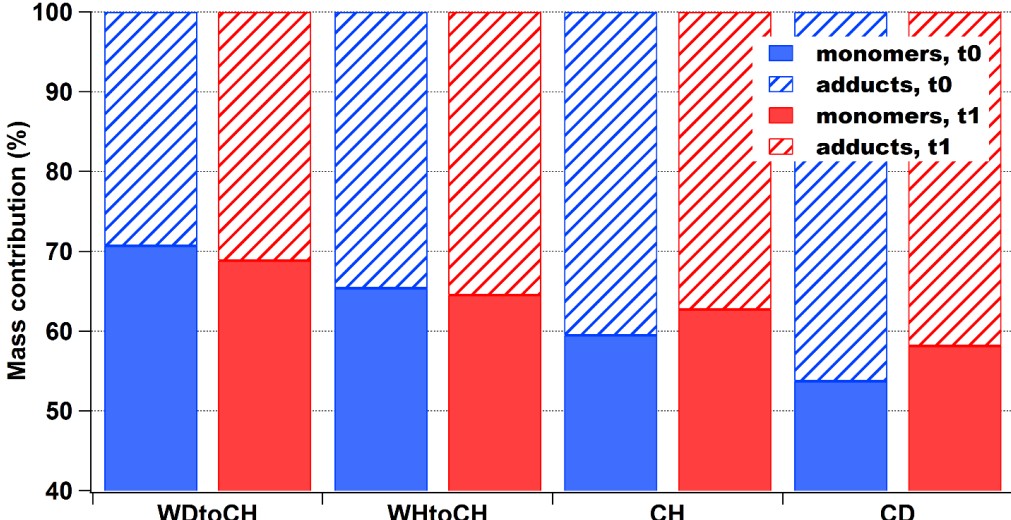

**Figure 4.** Relative mass contributions of monomers and adducts at t0 (blue) and t1 (red).





655

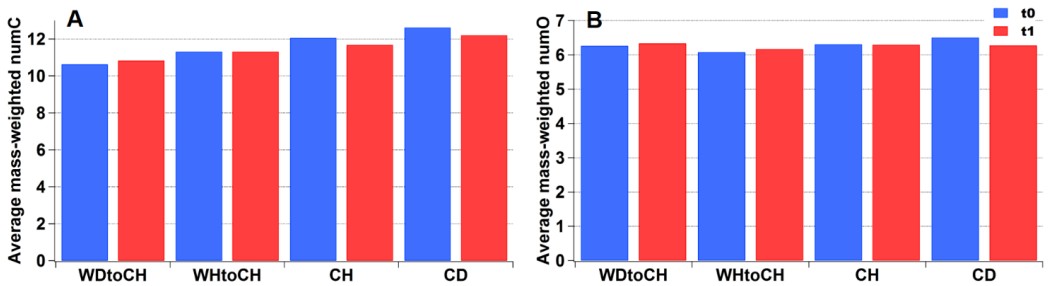

**Figure 5.** (A) Average mass-weighted number of carbon atoms (numC) and (B) oxygen atoms (numO) at t0 (blue) and t1 (red).



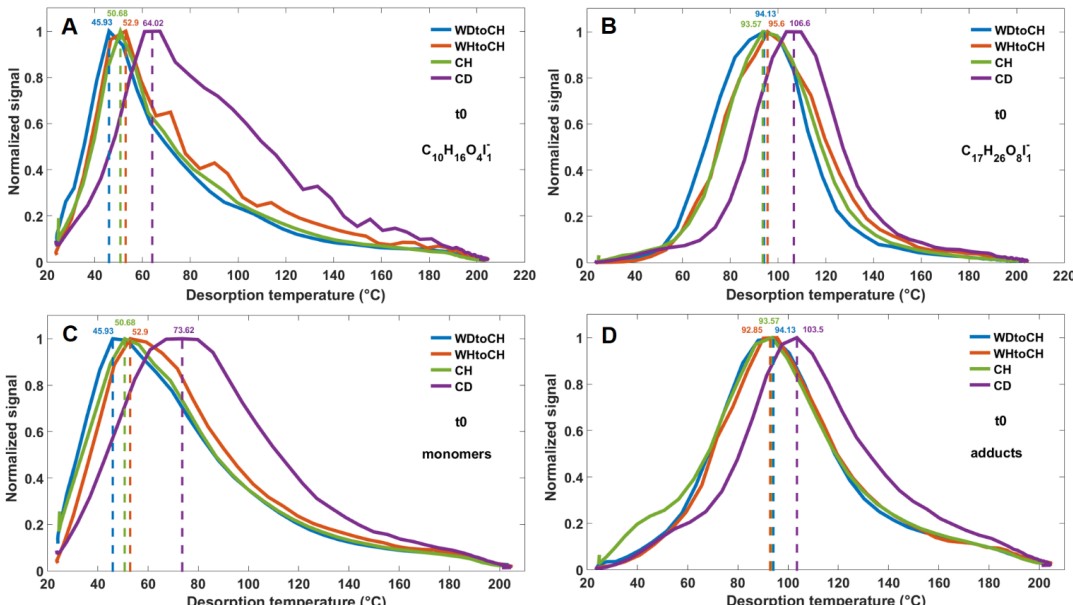

**Figure 6.** Thermograms of a monomer, $C_{10}H_{16}O_4$ (A) and an adduct, $C_{17}H_{26}O_8$ (B) both clustered with $I^-$ at t0, and sum thermograms of monomers (C) and adducts (D) at t0. Dashed lines refer to the corresponding $T_{max}$.





660

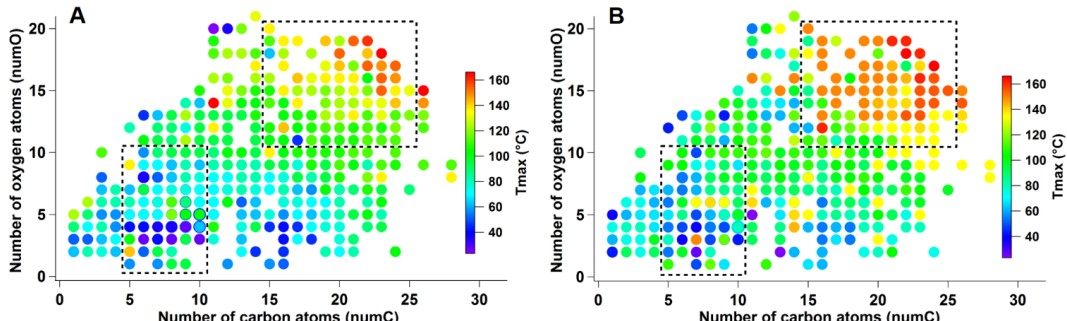

**Figure 7.** $T_{max}$ distribution for individual CHOI compounds of WDtoCH (A) and CD (B) experiments at t0 according to number of oxygen atoms (numO) vs Number of carbon atoms (numC). Dashed boxes specify the compounds with nominal molecular formulae $C_{5-10}H_yO_{1-10}I^-$ and $C_{15-25}H_yO_{11-20}I^-$ which had bigger $T_{max}$ differences.



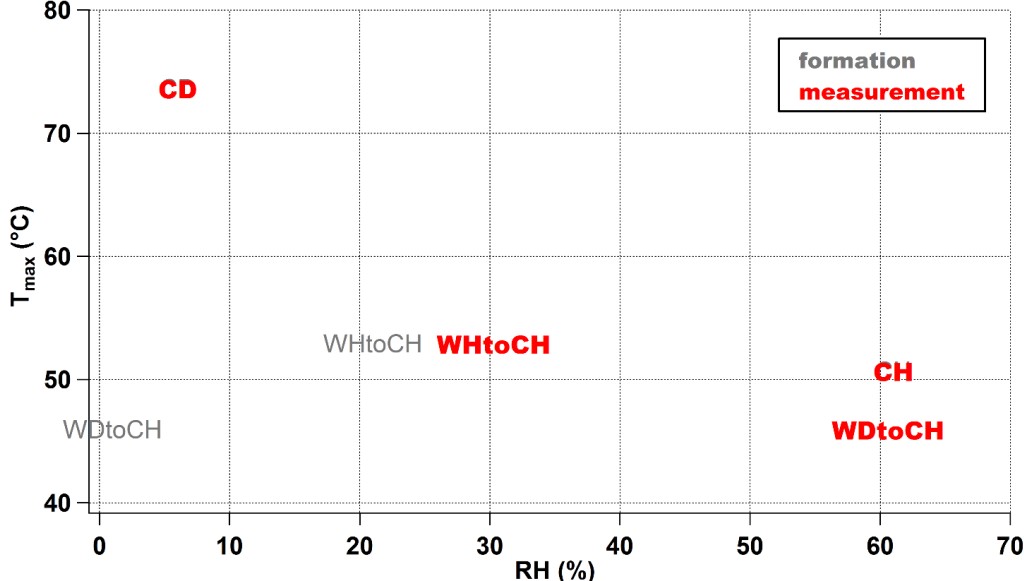

665 **Figure 8.** Relationship of $RH_{formation}$ (gray), $RH_{measurement}$ (red) and $T_{max}$ of all CHOI monomer compounds for four experiments at t0.