# Peer review of "α-pinene secondary organic aerosol at low temperature: Chemical composition and implications for particle viscosity"

_Atmospheric Chemistry and Physics, 2017_

## Referee Comment (RC1) · Anonymous Referee #1 · 26 Sep 2017

Review of ACP-2017-793 - $\alpha$-pinene secondary organic aerosol at low temperature: Chemical composition and implications for particle viscosity

This is a concise manuscript that presents necessary research into the Tmax interpretation from particle measurements utilising the FIGAERO ToF CIMS. I feel the results represent the work and analysis accurately, although I would suggest the following 2 points should be further investigated, as well as additional comments below.

1) I feel the application of a maximum calibration is inadequate for this work, or should be further investigated. The compounds measured are neither known to possess the maximum sensitivity nor in any way validated by calibrations presented. The further

discrepancy with minimal mass loadings between the AMS OA and CIMS CHOI further support inaccuracies in this determination. N2O5 has been determined to possess the maximum sensitivity in the CIMS (Lopez-Hilfiker et al., 2015) which possesses no similar functionality or mass. Calibration of a compound which can represent the products is necessary to validate any quantification from the CIMS measurements.

2) A variation of RH and temperature is interpreted to change the Tmax of thermograms from CIMS. It would be possible to calibrate for inactive or isotopically labelled species in varying temperatures and RHs to isolate these variables and prove via independent tests that they are indeed responsible for variations in Tmax through changes in viscosity.

Line 13 – Change comma to semi-colon: Karlsruhe Institute of Technology; the Aerosol

Line 21 – insert "the" with "the filter for gases or utilising the filter for gases"

Line 45 – replace like with "such as"

Line 46 to 48 – Very short sentences. I advise to rephrase or extend

Line 56 – VOC should be VOCs

Line 60 - Give a range of upper contribution of SOA from monoterpene oxidation products

Line 63 – Superscript radical charge

Line 66 – use O3 instead of ozone as already defined previously or maintain ozone throughout

Line 77 – "SOA is a highly dynamic system" – It does have a highly dynamic system but is phrased badly

Line 85 - I would restructure sentence not to start with "E.g."

Line 125 – Temperature should be temperatures

Line 130 – Instead of "right next to" describe distance or rephrase

Line 159 – Relative humidity (RH) has been defined a number of times in the paper so the acronym can be used throughout rather than repetition

Line 177 – Can a correlation be quantified between AMS and SMPS?

Line 179 - 180 Capitalise all or none of the letters used for the instrument acronym

Line 181 – Restructure sentence

Line 191 – Why was the maximum sensitivity attributed to these products? Furthermore, was a calibration performed to define this or merely taken from another CIMS sensitivity?

Line 192 – Were any backgrounds performed with no alpha pinene and varying ozone concentrations to quantify background oxidation of the chambers?

Line 195 – How was this calculated- can this temperature account for change in properties of deposited compounds on the filter? Can it be described how this was accounted in the interpretation of the results?

Line 205 – Please provide a validation as to why the measurements were not wall loss corrected and why this is interpreted as not necessary

Line 213 to 214 – The AMS observes a lower limit range of total organic mass (67.5) than the CIMS (97.8), which I cannot see possible. I disagree with applying a maximum sensitivity, which I describe below, but also this surely should then result in a minimal concentration. Here it seems the CIMS can overestimate the low concentrations and underestimate/represent the high concentrations. The selective ionisation and change in sensitivity, as stated on line 221, would make the CIMS mass always less than the AMS as the maximum sensitivity is applied.

Line 224 – There must be more validation of applying the maximum sensitivity to all products. I would like to see a calibration of an available compound which is of similar

mass or structure to that of the products observed. The maximum sensitivity is usually applied to halogens or non-collision limited species. None of this has bene exemplified here and I would therefore guess that this plays a much larger error in the concentration reported from the CIMS.

Line 240 – The loss due to evaporation could be quantified to a more accurate extent by running a double filter measurement by placing a filter further upstream of the current collection point. By performing a collection and desorbtion with two and one filters in line, some state of loss due could be accounted for.

Line 256 – Change biggest to largest or highest

Line 272 – Can error bars be added to this to illustrate a significant change between experiments

Line 294 – Thermograms has already defined, as well as Tmax

Line 370 – I agree with all interpretation with respect to Tmax variation, although I would like to see this determined under controlled conditions. Would it not be possible to calibrate for a compound, for instance a C10 deuterated carboxylic acid loaded into the chamber with varying mass loadings, RH and temperature. I feel there could be more work after the campaign on repeating Tmax variation due to the factors determined to be significant as found by the chamber experiments.
* * *

---

## Short Comment (SC1) · 10 Oct 2017

General comments:

My comments mainly relate to the chemical characterization of alpha-pinene SOA compounds, which could be elaborated in this manuscript. I found it interesting to see that the monomer $C_{10}H_{16}O_4$ and the dimer $C_{17}H_{26}O_8$ are major components of the alpha-pinene SOA system, and that the ratio monomers/oligomers is enhanced at the lower temperature.

Specific comments:

[Figure]

Line 68 - Introduction: The authors write: "The molecular formulae of organic species accounting for $\sim$58$-$72 % of SOA mass from $\alpha$-pinene ozonolysis have been identified, and can largely be grouped into monomers (C8$-$10H12$-$16O3$-$6, oxidation products from one $\alpha$-pinene molecule) and dimers (C14$-$19H24$-$28O5$-$9, oxidation products from two $\alpha$-pinene molecules) (Zhang et al., 2015)."

Since later in the manuscript results for the MW 358 dimer (C17H26O8) are selected, it would be worthwhile to also mention that major dimers of the alpha-pinene SOA system have been structurally elucidated. I suggest to add the following sentence: "Major dimers of the alpha-pinene SOA system have been structurally elucidated as a cis-pinyl-diaterpenyl ester (C17H26O8; MW 358) (Yasmeen et al., 2010) and a cis-pinyl-hydroxypinonyl ester (C19H28O7; MW 368) (Müller et al., 2008)."

Line 302: The authors write: "Examples of the thermograms of a monomer (C10H16O4, molecular formula corresponding to hydroxy-pinonic acid), and an adduct (C17H26O8, molecular formula identified in SOA from $\alpha$-pinene ozonolysis by e.g. Zhang et al., 2015; Mohr et al., 2017) both clustered with I- at t0 are shown in Fig. 6A$-$B."

With regard to the monomer C10H16O4, it would be worthwhile to consult the recent article by Zhang et al. (2017). These authors have studied HOMs in the alpha-pinene ozonolysis system and provided evidence for the formation of isomeric hydroxypinonic acids, with the hydroxy group at different positions of the pinonic acid skeleton, i.e., the 7- and 5-positions.

With regard to the C17H26O8 compound, it would be relevant to also mention the chemical structure. I suggest to do this as follows: ".... and a diester [C17H26O8, molecular structure identified in SOA from $\alpha$-pinene ozonolysis as a cis-pinyl-diaterpenyl ester (Yasmeen et al., 2010) and molecular formula identified in SOA from $\alpha$-pinene ozonolysis by e.g. Zhang et al., 2015; Mohr et al., 2017] both clustered ......"

Line 308: The authors write: "For experiment CD, the C10H16O4I1 - thermograms exhibited a multi-modal shape, indicative of contributions from isomers having different vapor pressures, or thermal decomposition of larger molecules. "

As already mentioned above, there is indeed evidence for different isomeric hydroxypinonic acids (Zhang et al., 2017), but decomposition of dimers is also a possibility. It is noted that the MW 368 diester, elucidated as a cis-pinyl-hydroxypinonyl, has a cis-pinic and 7-hydroxypinonic acid residue. In this respect, it would be interesting to examine the thermogram of cis-pinic acid (MW 186). Furthermore, it would also be worthwhile to examine the thermogram of terpenylic acid (MW 172), which could result from the degradation of the MW 358 diester.

Line 410: It is not clear what the authors mean by "adduct". I think they mean "oligomer", which is chemically more correct. The dimeric esters with MW 358 and 368, for example, are covalent dimers.

References:

Müller, L., Reinnig, M.-C., Warnke, J., and Hoffmann, Th.: Unambiguous identification of esters as oligomers in secondary organic aerosol formed from cyclohexene and cyclohexene/$\alpha$-pinene ozonolysis, Atmos. Chem. Phys., 8, 1423–1433, doi:10.5194/acp-8-1423-2008, 2008.

Yasmeen, F., Vermeylen, R., Szmigielski, R., Iinuma, Y., Böge, O., Herrmann, H., W. Maenhaut, W., and Claeys, M.: Terpenylic acid and related compounds: precursors for dimers in secondary organic aerosol from the ozonolysis of $\alpha$- and $\beta$-pinene, Atmos. Chem. Phys. 10, 9383–9392, doi:10.5194/acp-10-9383-2010, 2010.

Zhang, X., Lambe, A. T., Upshur, M. A., Brooks, W. A., Gray BeÌĄ, A., Thomson, R. J., Geiger, F. M., Surratt, J. D., Zhang, Z., Gold, A., Graf, S., Cubison, M. J., Groessl, M., Jayne, J. T., Worsnop, D. R., and Canagaratna, M. R.: Highly oxygenated multifunctional compounds in $\alpha$‑pinene secondary organic aerosol, Environ. Sci. Technol.

51, 5932–5940, 2017.

---

## Referee Comment (RC2) · Anonymous Referee #2 · 2 Nov 2017

General comment:

The authors discuss the formation of α-pinene secondary organic aerosol within two environmental chambers and its chemical-physical characterisation. Two different kinds of experiments are described and they are meant to mimic different possible formation and evolution conditions of SOA in the atmosphere, at different temperatures and relative humidities. The formed aerosol was characterised with a series of different analytical techniques, that allow the measurement of size distribution of particles, chemical composition and degree of oligomerisation, desorption temperatures. Generally speaking, the formation of SOA at low temperature has been very little studied in

the laboratory, because of evident experimental challenges that such measurements can present. I think that the potentiality of the authors' experimental setup in this respect is really interesting; the coupling of the two different types of experiments that they performed with the extensive chemical-physical characterisation of the formed $\alpha$-pinene SOA provides some interesting and valuable insights in this relatively little investigated field of atmospheric aerosols research. That said, I find there are a few points within the manuscript needs to be clarified, especially in the final discussion that the authors present at the end of the paper, where the inferred viscosity of the formed SOA is discussed. If the authors can address my points and questions below, I would recommend the publication of the manuscript in Atmospheric Chemistry and Physics.

Specific comments:

Line 59-60: "The fraction of total SOA mass from monoterpene oxidation products is estimated to be ∼15 % globally, and can be higher in 60 some regions (e.g. in the boreal forest) (Heald et al., 2008)". I think it would be worth mentioning here up to what kind of values the SOA from the oxidation of monoterpenes is estimated.

Line 123: "Here, we discuss a subset of the SOA15 dataset". This is not clear, do the authors mean that in this paper they are discussing a subset of 15 dataset, or a subset of a total of 15 experiments? I think it would be worth adding some information in the manuscript on how many experiments of each kind are being discussed, possibly in Table 1 where all the different conditions for all the different types of experiments are summarised.

Line 131: "was used to prepare SOA particles in a reproducible manner". Does this statement refer to the previous paper by Möhler et al. (2008) or is there data from this work that support this? Please add either the appropriate literature reference or report some data in support of this statement in the Supplement.

Line 144: "a fraction of the SOA particles was then transferred to the dark AIDA chamber". Could please the authors provide a little bit more detail on how this transfer is

done?

Line 147: "Figure 2", and Line 119. In panel A, which refers to the type 1 experiments, $\alpha$-pinene was added over two separate steps. Could please the authors mention why? I think this is a quite important point, since in the discussion of the results the concentration of $\alpha$-pinene and of SOA particles in the different kinds of experiments is often mentioned as a variable that influences the resulting chemical-physical properties of the produced aerosol (for example at Lines 283-291).

Lines 155-157: Could the authors please mention how this 5% uncertainty in the quantification of the water vapours and the uncertainty of the dew point hygrometer reflects on the final uncertainty on the determined RH value?

Line 176-177: "An AMS collection efficiency (CE) of $0.4-0.5$ was used, except for the CH experiment where CE was 0.7, likely due to higher particle water content". What was this decision based on? Could the authors please explain this a little more in detail or provide a literature reference to support it?

Line 193: "For each experiment, backgrounds were determined by sampling from the AIDA chamber before adding any precursor gases". If the authors want to mention this aspect of their experimental procedure, which I think is quite important, could they please indicate what they measured on average in the background or give a literature reference if such measurements were previously published? Or provide some data about this in the Supplement?

Lines 283-291: I agree with the general conclusion of this paragraph, the authors state that from these results the importance of the experimental conditions when interpreting laboratory data is fundamental. As I mentioned in the comment on Line 147, I believe this discussion would benefit from the authors' explanation of why the concentration $\alpha$-pinene was kept that much higher for type 1 experiments. If the two sets of experiments were performed at the same (or at a more comparable) VOC concentration this would not be a variable anymore and they would be able to completely discriminate the effects

of $\alpha$-pinene concentration from the effects of temperature on the chemical-physical properties of the formed SOA. Are the authors thinking of working in this direction?

Line 299 and following discussion: "Here we show that thermograms may also be used for qualitative information on particle viscosity - My biggest concern with the author's claim that they can get some information on the particles' viscosity from thermograms regards what is actually the effect of viscosity at the high temperatures at which the thermograms are taken. The authors say: "at the higher temperatures where adducts desorb, particle matrix effects may become less important" (Lines 324-325), including particles viscosity. By heating up the deposited particles, the viscosity of the particles is going to change (decrease); even at the relatively low Tmax they measured at about 45 °C, is the viscosity of the particles still going to play a role in the rate of diffusion of the desorbed compounds? And what about when the temperature is even further increased? Is there a way the authors can support these claims more strongly? - Line 365: "Calculated particle water content derived from AMS measurements is prone to large uncertainties (Engelhart et al., 2011); we observed a qualitative positive correlation with RH (data not shown)". If the authors want to mention this positive correlation I think it would be appropriate to show it in the Supplement, even if the water content quantification is characterised by a large uncertainty as they mention. - Line 376: "The dependency of Tmax on filter mass loading was not linear, and for our FIGAERO reached a plateau at mass loadings of 2−4 $\mu$g.". Looking at the data reported in the Supplement in Figure S6 panels B and D, I do not think it is true that Tmax plateaus for CD experiments. Both the shape of the thermograms and the Tmax value change with different mass loading. This could mean that all the previous discussion on the CD experiments could be affected by the choice of sampling times (Table S1). Could please the authors comment on this aspect and rephrase this statement?

Line 380: "In this study, $\alpha$-pinene SOA physicochemical properties such as chemical composition, phase state, and viscosity were investigated". I would tone this down and rephrase this because of the very uncertain link between what is observed in the ther-

mograms and the phase state/viscosity of the particles. Some aspects were actually investigated (size distributions, chemical composition, degree of oligomerisation, etc.) but the phase state/viscosity of the particles can just be supposed.

Technical comments:

Line 57: change "24.8 % mass contribution to global monoterpene emissions" to "24.8 % mass contribution to global monoterpenes emissions".

Line 183, 269 and 307: "Table 2". There is no Table 2 in the main manuscript, could please the authors double check this reference?

Figures 4,5, S2, S5 and 7: It is a little confusing for the reader having the results for type 2 experiments often displayed before type 1 experiment. I think it would help the reader if the sequence of the data displayed in the figures reflected the sequence with which the different kinds of experiments are presented in the abstract and in section 2.1. Possibly adding a label "Type 1 (or 2)" would help the reader, too.

Figures S3 and S4: I don't think these figures are referred to at any point in the manuscript. If this is the case, could please the authors either add references to these figures in the main manuscript or add some more context to them in the Supplement?

---

## Author Comment (AC1) · 21 Dec 2017

*Responses to reviewers' comments for manuscript*

**α-pinene secondary organic aerosol at low temperature: Chemical composition and implications for particle viscosity**

Wei Huang[1,2], Harald Saathoff[1], Aki Pajunoja[3], Xiaoli Shen[1,2], Karl-Heinz Naumann[1], Robert Wagner[1], Annele Virtanen[3], Thomas Leisner[1], Claudia Mohr[1,4,*]

[1]Institute of Meteorology and Climate Research, Karlsruhe Institute of Technology, Eggenstein-Leopoldshafen, 76344, Germany

[2]Institute of Geography and Geoecology, Karlsruhe Institute of Technology, Karlsruhe, 76131, Germany

[3]Department of Applied Physics, University of Eastern Finland, Kuopio, 80101, Finland

[4]Department of Environmental Science and Analytical Chemistry, Stockholm University, Stockholm, 11418, Sweden

*Correspondence to: C. Mohr (claudia.mohr@aces.su.se)

*We thank the Reviewers for their evaluation of the manuscript. Replies to the individual comments are directly added below them in italics. Please note that only references that are part of the replies to the comments are listed in the bibliography at the end of this document. References in copied text excerpts from the manuscript are not included in the bibliography. Page and line numbers refer to the original manuscript text.*

**Reviewer #1** *(responses in italics)*

This is a concise manuscript that presents necessary research into the Tmax interpretation from particle measurements utilising the FIGAERO ToF CIMS. I feel the results represent the work and analysis accurately, although I would suggest the following 2 points should be further investigated, as well as additional comments below.

1) I feel the application of a maximum calibration is inadequate for this work, or should be further investigated. The compounds measured are neither known to possess the

maximum sensitivity nor in any way validated by calibrations presented. The further discrepancy with minimal mass loadings between the AMS OA and CIMS CHOI further support inaccuracies in this determination. N2O5 has been determined to possess the maximum sensitivity in the CIMS (Lopez-Hilfiker et al., 2015) which possesses no similar functionality or mass. Calibration of a compound which can represent the products is necessary to validate any quantification from the CIMS measurements.

*It is not the goal of this manuscript to quantify total mass loadings, but to discuss relative changes in mass spectra and maximum desorption temperatures of individual compounds due to varying temperature and relative humidity (RH) conditions. Our main results are therefore largely independent of absolute mass concentrations. We clearly state in the manuscript (paragraph 3.1) that the FIGAERO-CIMS mass loadings represent a lower limit. Having said this, the ratios of the FIGAERO-CIMS mass concentrations to total organic particle mass measured by AMS we present here are well in line with earlier determinations of this ratio for α-pinene SOA from the laboratory or field, where the collisional limit sensitivity was applied to FIGAERO-CIMS data (e.g. Lopez-Hilfiker et al., 2016; Mohr et al., 2017). In addition, Lee et al. (2014) showed that the sensitivity to individual organic acids of the FIGAERO-CIMS using I⁻ as reagent ion approaches the value of the collisional limit for larger molecules (m/Q 300 and larger), which make up the major fraction of the signal in our mass spectra.*

2) A variation of RH and temperature is interpreted to change the Tmax of thermograms from CIMS. It would be possible to calibrate for inactive or isotopically labelled species in varying temperatures and RHs to isolate these variables and prove via independent tests that they are indeed responsible for variations in Tmax through changes in viscosity.

*Earlier studies have shown that CIMS thermograms and corresponding Tmax are very reproducible under stable conditions (Lopez-Hilfiker et al., 2014). We therefore expect the observed variations in Tmax to result from the changes in particle viscosity. We agree with the reviewer that the separation of the temperature and RH variables is very important. As we explain in lines 343–348 (section 3.3) of our manuscript, the temperature in AIDA should not have an effect on the variation of Tmax for the different experiments, as it was always 223 K: "Since the temperature was 223 K in AIDA for*

*all experiments discussed here, the observed differences in Tmax, and presumed viscosity, cannot be directly explained by differences in temperature. In addition, during desorption of compounds with the FIGAERO, particles are actively heated (with heat transfer assumed to be immediate), and are not evaporating under equilibrium conditions. Presumed variations in particle viscosity based on observed variations in Tmax must therefore be due to variations in particle chemical composition, and/or RH differences". However, temperature may have played an indirect role for Tmax variations: The temperature during SOA formation in our experiments was either 223 or 296 K, which may have influenced the chemistry (chemical composition, oligomerization degree, etc.), and thus particle viscosity. We state this e.g. in lines 289–290: "[…] the differences observed in mass spectral patterns between the two types of experiments are a consequence of both temperature and precursor concentration differences." The influence of RH on Tmax variations is explained in the manuscript in e.g. lines 357–358, lines 362–365, and Fig. 8.*

Line 13 – Change comma to semi-colon: Karlsruhe Institute of Technology; the Aerosol

*Done.*

Line 21 – insert "the" with "the filter for gases or utilising the filter for gases"

*We replaced "with" by "coupled to a".*

Line 45 – replace like with "such as"

*Done.*

Line 46 to 48 – Very short sentences. I advise to rephrase or extend

*The two sentences were combined to one.*

Line 56 – VOC should be VOCs

*Corrected throughout the manuscript.*

Line 60 - Give a range of upper contribution of SOA from monoterpene oxidation products

*Due to a lack of published numbers, the sentence was changed to : "SOA from monoterpenes is very important in the boreal regions in summertime, and the fraction*

*of total SOA mass from monoterpene oxidation products is estimated to be ~15 % globally (Heald et al., 2008)."*

Line 63 – Superscript radical charge

*For clarification, the sentence was changed to: "The reactions of α-pinene with $O_3$, and the radicals OH and $NO_3$ lead to […]".*

Line 66 – use O3 instead of ozone as already defined previously or maintain ozone throughout

*$O_3$ is used throughout the manuscript.*

Line 77 – "SOA is a highly dynamic system" – It does have a highly dynamic system

but is phrased badly

*Sentence rephrased as following: "SOA is highly dynamic and continually evolves in the atmosphere, becoming increasingly oxidized, less volatile, and more hygroscopic […]".*

Line 85 - I would restructure sentence not to start with "E.g."

*Sentence rephrased as following: "Water diffusion coefficients in the water-soluble fraction of α-pinene SOA, were e.g. measured for temperatures between 240 and 280 K."*

Line 125 – Temperature should be temperatures

*Corrected.*

Line 130 – Instead of "right next to" describe distance or rephrase

*Sentence rephrased as following: "The APC (Aerosol Preparation and Characterization) chamber (Möhler et al., 2008) is a 3.7 $m^3$ sized stainless steel vessel, situated at a distance of 3 m from AIDA and connected to it by a 7 m stainless steel tube of 24 mm inner diameter."*

Line 159 – Relative humidity (RH) has been defined a number of times in the paper so the acronym can be used throughout rather than repetition

*Corrected throughout the manuscript.*

Line 177 – Can a correlation be quantified between AMS and SMPS?

*Correlation coefficients and slopes were calculated for the experiments shown here, and the values were added to the manuscript as following: "(slopes are between 0.87−1.04 except for CD experiment (2.2) possibly due to the lower transmission efficiency in the aerodynamic lens of the AMS for sub-100-nm particles, Pearson's correlation coefficients are between 0.87−0.98 for the experiments presented here)."*

Line 179 - 180 Capitalise all or none of the letters used for the instrument acronym

*Corrected as suggested.*

Line 181 – Restructure sentence

*R: Sentence restructured as following: "During the gas phase measurement, gases were sampled via a Fluorinated Ethylene Propylene (FEP) tube of 0.83 m length while particles were simultaneously collected on a Teflon (Polytetrafluoroethylene, PTFE) filter [...]".*

Line 191 – Why was the maximum sensitivity attributed to these products? Furthermore, was a calibration performed to define this or merely taken from another CIMS sensitivity?

*Compare to the response to the first comment.*

Line 192 – Were any backgrounds performed with no alpha pinene and varying ozone concentrations to quantify background oxidation of the chambers?

*Each experiment was started by measuring background air and background particles in AIDA before addition of any trace gases. For type 2 experiments, we did not observe a significant background before the SOA transfer from the APC chamber and initial particle number concentrations were typically below 1 $cm^{-3}$. For type 1 experiments, $O_3$ was usually added first and α-pinene was added last. We observed only a small increases in both gas mixing ratio and particle mass ($<0.01\mu g$ $m^{-3}$) after $O_3$ addition. However, these background concentrations were also negligible compared to the increase by the SOA generated (>1000 fold for particle mass). The following sentence was added in the manuscript: "[...] For type 2 experiments, backgrounds were negligible with initial particle number concentrations below 1 $cm^{-3}$. For type 1 experiments, we observed a small increase in both gas mixing ratio and particle mass*

*(<0.01 μg m$^{-3}$) after O$_3$ addition, which was subtracted from the mass loadings presented here. However, the background and the increase induced by O$_3$ addition were negligible compared to the increase by the SOA mass (>1000 fold for particle mass)."*

Line 195 – How was this calculated- can this temperature account for change in properties of deposited compounds on the filter? Can it be described how this was accounted in the interpretation of the results?

*The temperature increase for the particle inlet can be calculated as described by Fitzer and Fritz (1989) using the dimensions of the stainless steel tube and the insulation material Armaflex, the corresponding heat conductivities, and the flowrate through the sampling tube. As we state in lines 195–197, due to the possible increase in temperature of the sample during deposition, we cannot entirely rule out partial evaporation of water or semi-volatile organic compounds. It is therefore possible that in our particle phase data highly volatile compounds may be slightly underestimated. Since temperature and sampling conditions were kept the same for all four experiments discussed in the manuscript, we assume the artefacts to be similar and thus our main results not to be affected by them. The corresponding reference was added to the manuscript.*

Line 205 – Please provide a validation as to why the measurements were not wall loss corrected and why this is interpreted as not necessary

*A detailed characterization of wall losses in the AIDA aluminum chamber using empirical data and the COSIMA model (Naumann, 2003) is subject of an upcoming publication. In our manuscript we are mostly interested in the relation between particle chemical composition and maximum desorption temperature (Tmax), which is completely independent of wall losses.*

Line 213 to 214 – The AMS observes a lower limit range of total organic mass (67.5) than the CIMS (97.8), which I cannot see possible. I disagree with applying a maximum sensitivity, which I describe below, but also this surely should then result in a minimal concentration. Here it seems the CIMS can overestimate the low concentrations and underestimate/represent the high concentrations. The selective ionisation and change in sensitivity, as stated on line 221, would make the CIMS mass always less than the AMS as the maximum sensitivity is applied.

*As we state in lines 225–228, the reason why the AMS observes lower organic mass concentrations (67.5 µg m$^{-3}$) than the CIMS (97.8 µg m$^{-3}$) for type 1 experiments is the reduced transmission of the aerodynamic lens for sub-100-nm particles (see the beginning of the CH experiment at time point t0 in Fig. 2C). After the particles have grown in size (Fig. 2D) the AMS observes more total organic mass (440.1 µg m$^{-3}$) than the CIMS (247.6 µg m$^{-3}$) due to selective ionization (the numbers were stated in lines 213–214). Please see our response to the first comment.*

Line 224 – There must be more validation of applying the maximum sensitivity to all products. I would like to see a calibration of an available compound which is of similar mass or structure to that of the products observed. The maximum sensitivity is usually applied to halogens or non-collision limited species. None of this has bene exemplified here and I would therefore guess that this plays a much larger error in the concentration reported from the CIMS.

*Please see our response to the first comment. The maximum sensitivity has been used for organic compounds measured by FIGAERO-CIMS using Iodide earlier (Mohr et al., 2017). In addition, Lopez-Hilfiker et al. (2015) state that " [...] the maximum sensitivity provides a critical constraint on the sensitivity of a ToF-CIMS [using Iodide] towards a wide suite of routinely detected multifunctional organic molecules for which no calibration standards exist."*

Line 240 – The loss due to evaporation could be quantified to a more accurate extent by running a double filter measurement by placing a filter further upstream of the current collection point. By performing a collection and desorbtion with two and one filters in line, some state of loss due could be accounted for.

*We normally use a second filter upstream of the filter used for particle collection for particle zeroing, in order to know the contribution of gases potentially absorbing on the filter to the particle signal. For a more quantitative assessment of potential evaporation losses, a comparison of particle chemical composition (mass spectra) as a function of deposition time at the temperature conditions of our experiments would need to be performed, which however lies outside the scope of this paper.*

Line 256 – Change biggest to largest or highest

*Done.*

Line 272 – Can error bars be added to this to illustrate a significant change between experiments

*The average mass-weighted numC or numO values were calculated based on the sum of the product of carbon number or oxygen number of each compound multiplying the corresponding mass contribution of each compound. The only error bar that can be induced in this calculation is the uncertainty of the signal for each compound in the mass spectra. Assuming an uncertainty of 35 % for CIMS (Mohr et al., 2017), error bars were calculated and added for the average mass-weighted numC and numO as well as the mass contributions and concentrations for monomers and adducts in Figure 4, Figure 5, and Figure S2 respectively.*

Line 294 – Thermograms has already defined, as well as Tmax

*The repetitive definitions of thermograms and Tmax were removed.*

Line 370 – I agree with all interpretation with respect to Tmax variation, although I would like to see this determined under controlled conditions. Would it not be possible to calibrate for a compound, for instance a C10 deuterated carboxylic acid loaded into the chamber with varying mass loadings, RH and temperature. I feel there could be more work after the campaign on repeating Tmax variation due to the factors determined to be significant as found by the chamber experiments.

*We thank the reviewer for this suggestion, and we agree that more experiments might be needed in the future to further investigate particle viscosity and desorption behavior. However, using a single compound might not necessarily result in desired further clarification. As we state at several occasions in the manuscript (e.g. lines 335–338), our results show that Tmax is not purely a function of the compounds' vapor pressures or volatilities, but is influenced by diffusion limitations within the particles (particle viscosity) (Vaden et al., 2011; Yli-Juuti et al., 2017), interactions between particles deposited on the filter (particle matrix), and/or particle mass on the filter. Just adding a single compound to the AIDA chamber would not take the complex particle matrix resulting from SOA formation into account.*

**Reviewer #2** *(responses in italics)*

General comment:

The authors discuss the formation of α-pinene secondary organic aerosol within two environmental chambers and its chemical-physical characterisation. Two different kinds of experiments are described and they are meant to mimic different possible formation and evolution conditions of SOA in the atmosphere, at different temperatures and relative humidities. The formed aerosol was characterised with a series of different analytical techniques, that allow the measurement of size distribution of particles, chemical composition and degree of oligomerisation, desorption temperatures. Generally speaking, the formation of SOA at low temperature has been very little studied in the laboratory, because of evident experimental challenges that such measurements can present. I think that the potentiality of the authors' experimental setup in this respect is really interesting; the coupling of the two different types of experiments that they performed with the extensive chemical-physical characterisation of the formed α-pinene SOA provides some interesting and valuable insights in this relatively little investigated field of atmospheric aerosols research. That said, I find there are a few points within the manuscript needs to be clarified, especially in the final discussion that the authors present at the end of the paper, where the inferred viscosity of the formed SOA is discussed. If the authors can address my points and questions below, I would recommend the publication of the manuscript in Atmospheric Chemistry and Physics.

Specific comments:

Line 59-60: "The fraction of total SOA mass from monoterpene oxidation products is estimated to be ~15 % globally, and can be higher in 60 some regions (e.g. in the boreal forest) (Heald et al., 2008)". I think it would be worth mentioning here up to what kind of values the SOA from the oxidation of monoterpenes is estimated.

*Due to a lack of published numbers, the sentence was changed to : "SOA from monoterpenes is very important in the boreal regions in summertime, and the fraction of total SOA mass from monoterpene oxidation products is estimated to be ~15 % globally (Heald et al., 2008)."*

Line 123: "Here, we discuss a subset of the SOA15 dataset". This is not clear, do the authors mean that in this paper they are discussing a subset of 15 dataset, or a subset of a total of 15 experiments? I think it would be worth adding some information in the

manuscript on how many experiments of each kind are being discussed, possibly in Table 1 where all the different conditions for all the different types of experiments are summarised.

*For clarification, the corresponding sentence was changed to: "Here, we discuss a subset (Table 1) of the large dataset of the SOA15 campaign that is based on experiments [...]".*

Line 131: "was used to prepare SOA particles in a reproducible manner". Does this statement refer to the previous paper by Möhler et al. (2008) or is there data from this work that support this? Please add either the appropriate literature reference or report some data in support of this statement in the Supplement.

*The corresponding reference (Wagner et al., 2017) was added.*

Line 144: "a fraction of the SOA particles was then transferred to the dark AIDA chamber". Could please the authors provide a little bit more detail on how this transfer is done?

*The SOA transfer to AIDA was done using a 7 m stainless steel tube with 24 mm inner diameter (this information was added to the manuscript). The transfer of particles is described in lines 143–146: "After a residence time of 1−1.5 h in the APC chamber, its pressure was increased by 5 hPa compared to AIDA and a fraction of the SOA particles was then transferred to the dark AIDA chamber kept at 223 K, 61 % RH (WDtoCH) or 30 % RH (WHtoCH), respectively, resulting in the particle number concentrations ranging between 1500−2200 cm$^{-3}$ in the AIDA chamber."*

Line 147: "Figure 2", and Line 119. In panel A, which refers to the type 1 experiments, α-pinene was added over two separate steps. Could please the authors mention why? I think this is a quite important point, since in the discussion of the results the concentration of α-pinene and of SOA particles in the different kinds of experiments is often mentioned as a variable that influences the resulting chemical-physical properties of the produced aerosol (for example at Lines 283-291).

*CH and CD are both type 1 experiments. Only for the CH experiment, α-pinene was added over two separate steps to increase SOA mass. This however will not influence the relationship between particle chemical composition and Tmax distribution.*

Lines 155-157: Could the authors please mention how this 5% uncertainty in the quantification of the water vapours and the uncertainty of the dew point hygrometer reflects on the final uncertainty on the determined RH value?

*Our TDL hygrometer has an accuracy of ±5 % in the water vapor pressure and our dew point mirror (MBW373LX) has an accuracy of ±0.1 °C corresponding to ±1 % in the water vapor pressure. Typically, both instruments agree within ±2 %. For the determination of the relative humidity, the temperature uncertainty needs to be taken into consideration and this is ± 0.3 °C expressed as the temperature inhomogeneity in the AIDA chamber under static conditions. This corresponds to ±3 % uncertainty in the water vapor pressure. Combining the observed uncertainties for water vapor pressure and temperature results in an uncertainty of ±5 % for the relative humidity. The following sentence was added in the manuscript: "Temperature (T) in the AIDA chamber was measured by home-made thermocouples (NiCrNi) and home-made PT 100 temperature sensors with an accuracy of ±3 %, […] a dew point mirror hygrometer (MBW373LX, MBW Calibration Ltd.) with an accuracy of ±1 % traceable to different national metrology standards including Federal Institution of Physical Technology (PTB), National Physical Laboratory (NPL), Federal Office of Metrology and Surveying (BEV) and NIST. Both instruments agree within ±2 %. Relative humidity (RH) in the AIDA chamber was calculated using the measured water vapor concentrations and temperature based on the saturation water vapor pressures given by Murphy and Koop (2005), resulting in an accuracy of ±5 %."*

Line 176-177: "An AMS collection efficiency (CE) of 0.4-0.5 was used, except for the CH experiment where CE was 0.7, likely due to higher particle water content". What was this decision based on? Could the authors please explain this a little more in detail or provide a literature reference to support it?

*AMS CE is expressed by the product of 3 terms, transmission efficiency of the aerodynamic lens for spherical particles ($E_L$), the loss of transmission due to particle nonsphericity which causes the particle beam to broaden ($E_S$), and the efficiency with which a particle that impacts the vaporizer is detected (Eb) (Huffman et al., 2005). It is influenced by aerosol composition and sampling line relative humidity (Middlebrook et al., 2012). We expect higher CE with higher relative humidity in the sampling line due to reduced bouncing of particles from the vaporizer. A CE of ~0.5 has been found for many AMS from both field and laboratory studies (e.g. Aiken et al., 2009; Takegawa*

*et al., 2005; Middlebrook et al., 2012; Robinson et al., 2017; Matthew et al., 2008),
and comparison with ancillary data also indicate a CE close to 0.5 for our instrument.
The corresponding reference was added to the manuscript.*

Line 193: "For each experiment, backgrounds were determined by sampling from the AIDA chamber before adding any precursor gases". If the authors want to mention this aspect of their experimental procedure, which I think is quite important, could they please indicate what they measured on average in the background or give a literature reference if such measurements were previously published? Or provide some data about this in the Supplement?

*Each experiment was started by measuring background air and background particles in AIDA before addition of any trace gases. For type 2 experiments, we did not observe a significant background before the SOA transfer from the APC chamber and initial particle number concentrations were typically below 1 cm$^{-3}$. For type 1 experiments, O$_3$ was usually added first and α-pinene was added last. We observed only a small increases in both gas mixing ratio and particle mass (<0.01µg m$^{-3}$) after O$_3$ addition. However, these background concentrations were also negligible compared to the increase by the SOA generated (>1000 fold for particle mass). The following sentence was added in the manuscript: "[...] For type 2 experiments, backgrounds were negligible with initial particle number concentrations below 1 cm$^{-3}$. For type 1 experiments, we observed a small increase in both gas mixing ratio and particle mass (<0.01 µg m$^{-3}$) after O$_3$ addition, which was subtracted from the mass loadings presented here. However, the background and the increase induced by O$_3$ addition were negligible compared to the increase by the SOA mass (>1000 fold for particle mass)."*

Lines 283-291: I agree with the general conclusion of this paragraph, the authors state that from these results the importance of the experimental conditions when interpreting laboratory data is fundamental. As I mentioned in the comment on Line 147, I believe this discussion would benefit from the authors' explanation of why the concentration α-pinene was kept that much higher for type 1 experiments. If the two sets of experiments were performed at the same (or at a more comparable) VOC concentration this would not be a variable anymore and they would be able to completely discriminate the effects of α-pinene concentration from the effects of temperature on the chemical-physical properties of the formed SOA. Are the authors thinking of working in this direction?

*We would like to mention that the α-pinene concentration was kept the same within the same experiment type (i.e. type 1 or type 2), but there were differences in α-pinene concentrations between type 1 and type 2 experiments (concentrations were by a factor of 3 higher for type 2 experiments). We are aware of the influence of the differences in precursor concentrations (e.g. lines 241–243, 289–290). However, even within each type of experiment, we also observed differences in mass spectral patterns, size distributions, degree of oligomerization, thermograms, etc. Therefore, we could not completely discriminate the effects of α-pinene concentration from the effects of temperature on the chemical-physical properties of the formed SOA. However, for future studies precursor conditions would be ideally be kept constant.*

Line 299 and following discussion: "Here we show that thermograms may also be used for qualitative information on particle viscosity - My biggest concern with the author's claim that they can get some information on the particles' viscosity from thermograms regards what is actually the effect of viscosity at the high temperatures at which the thermograms are taken. The authors say: "at the higher temperatures where adducts desorb, particle matrix effects may become less important" (Lines 324-325), including particles viscosity. By heating up the deposited particles, the viscosity of the particles is going to change (decrease); even at the relatively low Tmax they measured at about 45 °C, is the viscosity of the particles still going to play a role in the rate of diffusion of the desorbed compounds? And what about when the temperature is even further increased? Is there a way the authors can support these claims more strongly?

*We are limited in our possibilities of heating the particles to more than 200 °C due to the material of the FIGAERO (Teflon), which starts to melt at ~240 °C. We therefore cannot extend our studies to higher temperatures, unfortunately. Our empirical data on varying Tmax for the same individual compound for different experimental conditions indicate that indeed that there must be factors other than the vapor pressure of the respective compound that influence Tmax, even at those elevated temperatures applied in the FIGAERO. However, in line with the reviewer´s comment, and as stated in lines 323–325 in the manuscript: "Variation in Tmax of the sum of CHOI compounds was larger for monomers (Fig. 6C) than for adducts (Fig. 6D). Monomers are thus the more important contributors to the shifts in Tmax, likely because at the higher temperatures where adducts desorb, particle matrix effects may become less important."*

Line 365: "Calculated particle water content derived from AMS measurements is prone

to large uncertainties (Engelhart et al., 2011); we observed a qualitative positive correlation with RH (data not shown)". If the authors want to mention this positive correlation I think it would be appropriate to show it in the Supplement, even if the water content quantification is characterised by a large uncertainty as they mention.

*We agree with the reviewer that mentioning particle water content in the manuscript would warrant to show the data in the Supplement. However, as we are not confident enough with the quality of that data we have decided to remove that information from the manuscript.*

Line 376: "The dependency of Tmax on filter mass loading was not linear, and for our FIGAERO reached a plateau at mass loadings of 2−4 µg.". Looking at the data reported in the Supplement in Figure S6 panels B and D, I do not think it is true that Tmax plateaus for CD experiments. Both the shape of the thermograms and the Tmax value change with different mass loading. This could mean that all the previous discussion on the CD experiments could be affected by the choice of sampling times (Table S1). Could please the authors comment on this aspect and rephrase this statement?

*Figure S6 shows the sum thermograms of all CHOI compounds with different sampling time (which also means different mass loadings here). As described in the manuscript (lines 325−326), the sum of thermograms and its Tmax is highly influenced by compounds with large signal, we therefore also show a box and whisker diagram of Tmax distribution for CHOI compounds, CHOI monomers and adducts in the Supplement (Figure S7). In this figure we can observe that beyond filter mass loadings of 2−4 µg the curves level off (saturation effect).*

Line 380: "In this study, α-pinene SOA physicochemical properties such as chemical composition, phase state, and viscosity were investigated". I would tone this down and rephrase this because of the very uncertain link between what is observed in the thermograms and the phase state/viscosity of the particles. Some aspects were actually investigated (size distributions, chemical composition, degree of oligomerisation, etc.) but the phase state/viscosity of the particles can just be supposed.

*The sentence was rephrased as following: "In this study, α-pinene SOA physicochemical properties such as chemical composition, size distributions, and degree of oligomerization were investigated at low temperature (223 K) and different relative humidity (RH) […]".*

Technical comments:

Line 57: change "24.8 % mass contribution to global monoterpene emissions" to "24.8 % mass contribution to global monoterpenes emissions".

*Corrected as suggested.*

Line 183, 269 and 307: "Table 2". There is no Table 2 in the main manuscript, could please the authors double check this reference?

*It is actually Table S1. Corrected throughout the manuscript.*

Figures 4,5, S2, S5 and 7: It is a little confusing for the reader having the results for type 2 experiments often displayed before type 1 experiment. I think it would help the reader if the sequence of the data displayed in the figures reflected the sequence with which the different kinds of experiments are presented in the abstract and in section 2.1. Possibly adding a label "Type 1 (or 2)" would help the reader, too.

*Labels were added in Figures 4, 5, S2, S5, 7 as suggested.*

Figures S3 and S4: I don't think these figures are referred to at any point in the manuscript. If this is the case, could please the authors either add references to these figures in the main manuscript or add some more context to them in the Supplement?

*Figures S3 and S4 are both referred in Section 3.3 in the manuscript (lines 305−306 for Figure S3 and lines 314−316 for Figure S4).*

**M. Claeys** *(responses in italics)*

General comments:

My comments mainly relate to the chemical characterization of alpha-pinene SOA compounds, which could be elaborated in this manuscript. I found it interesting to see that the monomer $C_{10}H_{16}O_4$ and the dimer $C_{17}H_{26}O_8$ are major components of the alpha-pinene SOA system, and that the ratio monomers/oligomers is enhanced at the lower temperature.

*We thank Prof. Claeys for her time in reading and commenting our manuscript. Replies*

*to the individual comments are directly added below them in italics.*

Specific comments:

Line 68 - Introduction: The authors write: "The molecular formulae of organic species accounting for ~58-72 % of SOA mass from α-pinene ozonolysis have been identified, and can largely be grouped into monomers (C8-10H12-16O3-6, oxidation products from one α-pinene molecule) and dimers (C14-19H24-28O5-9, oxidation products from two α-pinene molecules) (Zhang et al., 2015)."

Since later in the manuscript results for the MW 358 dimer (C17H26O8) are selected, it would be worthwhile to also mention that major dimers of the alpha-pinene SOA system have been structurally elucidated. I suggest to add the following sentence: "Major dimers of the alpha-pinene SOA system have been structurally elucidated as a cis-pinyl-diaterpenyl ester (C17H26O8; MW 358) (Yasmeen et al., 2010) and a cispinyl-hydroxypinonyl ester (C19H28O7; MW 368) (Müller et al., 2008)."

*Sentence added as suggested.*

Line 302: The authors write: "Examples of the thermograms of a monomer (C10H16O4, molecular formula corresponding to hydroxy-pinonic acid), and an adduct (C17H26O8, molecular formula identified in SOA from α-pinene ozonolysis by e.g. Zhang et al., 2015; Mohr et al., 2017) both clustered with I- at t0 are shown in Fig. 6A-B."

With regard to the monomer C10H16O4, it would be worthwhile to consult the recent article by Zhang et al. (2017). These authors have studied HOMs in the alpha-pinene ozonolysis system and provided evidence for the formation of isomeric hydroxypinonic acids, with the hydroxy group at different positions of the pinonic acid skeleton, i.e., the 7- and 5-positions.

With regard to the C17H26O8 compound, it would be relevant to also mention the chemical structure. I suggest to do this as follows: "…. and a diester [C17H26O8, molecular structure identified in SOA from α-pinene ozonolysis as a cispinyl-diaterpenyl ester (Yasmeen et al., 2010) and molecular formula identified in SOA from α-pinene ozonolysis by e.g. Zhang et al., 2015; Mohr et al., 2017] both clustered ……"

*Sentence corrected as suggested.*

Line 308: The authors write: "For experiment CD, the C10H16O4I1 – thermograms

exhibited a multi-modal shape, indicative of contributions from isomers having different vapor pressures, or thermal decomposition of larger molecules. "

As already mentioned above, there is indeed evidence for different isomeric hydroxypinonic acids (Zhang et al., 2017), but decomposition of dimers is also a possibility. It is noted that the MW 368 diester, elucidated as a cis-pinyl-hydroxypinonyl, has a cis-pinic and 7-hydroxypinonic acid residue. In this respect, it would be interesting to examine the thermogram of cis-pinic acid (MW 186). Furthermore, it would also be worthwhile to examine the thermogram of terpenylic acid (MW 172), which could result from the degradation of the MW 358 diester.

*According to Prof. Claeys suggestion we re-examined the data and indeed also found multi-mode thermograms for cis-pinic acid (MW 186) and terpenylic acid (MW 172). The following sentence was added in the manuscript: "[...] Different isomeric hydroxypinonic acids were found in α-pinene SOA (Zhang et al., 2017) and the decomposition of cis-pinyl-hydroxypinonyl diester could have a cis-pinic and 7-hydroxypinonic acid residue (Müller et al., 2008)."*

Line 410: It is not clear what the authors mean by "adduct". I think they mean "oligomer", which is chemically more correct. The dimeric esters with MW 358 and 368, for example, are covalent dimers.

*We changed the definition of adducts in lines 261–262 to: "[...] (this definition includes dimers, trimers, and oligomers in general)".*

***References:***

Aiken, A. C., Salcedo, D., Cubison, M. J., Huffman, J. A., DeCarlo, P. F., Ulbrich, I. M., Docherty, K. S., Sueper, D., Kimmel, J. R., Worsnop, D. R., Trimborn, A., Northway, M., Stone, E. A., Schauer, J. J., Volkamer, R. M., Fortner, E., de Foy, B., Wang, J., Laskin, A., Shutthanandan, V., Zheng, J., Zhang, R., Gaffney, J., Marley, N. A., Paredes-Miranda, G., Arnott, W. P., Molina, L. T., Sosa, G., and Jimenez, J. L.: Mexico city aerosol analysis during MILAGRO using high resolution aerosol mass spectrometry at the urban supersite (T0) - Part 1: Fine particle composition and organic source apportionment, Atmos Chem Phys, 9, 6633−6653, 2009.

Fitzer, E., and Fritz, W.: Technische Chemie, Third ed., Springer, Berlin, 140 pp., 1989.

Huffman, J. A., Jayne, J. T., Drewnick, F., Aiken, A. C., Onasch, T., Worsnop, D. R., and Jimenez, J. L.: Design, modeling, optimization, and experimental tests of a particle beam width probe for the aerodyne aerosol mass spectrometer, Aerosol Sci Tech, 39, 1143−1163, 2005.

Lee, B. H., Lopez-Hilfiker, F. D., Mohr, C., Kurtén, T., Worsnop, D. R., and Thornton, J. A.: An iodide-adduct high-resolution time-of-flight chemical-ionization mass spectrometer: Application to atmospheric inorganic and organic compounds, Environ Sci Technol, 48, 6309−6317, 2014.

Lopez-Hilfiker, F. D., Mohr, C., Ehn, M., Rubach, F., Kleist, E., Wildt, J., Mentel, T. F., Lutz, A., Hallquist, M., Worsnop, D., and Thornton, J. A.: A novel method for online analysis of gas and particle composition: description and evaluation of a Filter Inlet for Gases and AEROsols (FIGAERO), Atmos Meas Tech, 7, 983−1001, 2014.

Lopez-Hilfiker, F. D., Mohr, C., Ehn, M., Rubach, F., Kleist, E., Wildt, J., Mentel, T. F., Carrasquillo, A. J., Daumit, K. E., Hunter, J. F., Kroll, J. H., Worsnop, D. R., and Thornton, J. A.: Phase partitioning and volatility of secondary organic aerosol components formed from α-pinene ozonolysis and OH oxidation: the importance of accretion products and other low volatility compounds, Atmos Chem Phys, 15, 7765−7776, 2015.

Lopez-Hilfiker, F. D., Iyer, S., Mohr, C., Lee, B. H., D'Ambro, E. L., Kurtén, T., and Thornton, J. A.: Constraining the sensitivity of iodide adduct chemical ionization mass spectrometry to multifunctional organic molecules using the collision limit and thermodynamic stability of iodide ion adducts, Atmos Meas Tech, 9, 1505−1512, 2016.

Matthew, B. M., Middlebrook, A. M., and Onasch, T. B.: Collection efficiencies in an Aerodyne Aerosol Mass Spectrometer as a function of particle phase for laboratory generated aerosols, Aerosol Sci Tech, 42, 884−898, 2008.

Middlebrook, A. M., Bahreini, R., Jimenez, J. L., and Canagaratna, M. R.: Evaluation of composition-dependent collection efficiencies for the Aerodyne Aerosol Mass Spectrometer using field data, Aerosol Sci Tech, 46, 258−271, 2012.

Mohr, C., Lopez-Hilfiker, F. D., Yli-Juuti, T., Heitto, A., Lutz, A., Hallquist, M., D'Ambro, E. L., Rissanen, M. P., Hao, L. Q., Schobesberger, S., Kulmala, M., Mauldin, R. L., Makkonen, U., Sipilä, M., Petäjä, T., and Thornton, J. A.: Ambient observations of dimers from terpene oxidation in the gas phase: Implications for new particle formation and growth, Geophys Res Lett, 44, 2958−2966, 2017.

Naumann, K.-H.: COSIMA - a computer program simulating the dynamics of fractal aerosols, J Aerosol Sci, 34, 1371−1397, 2003.

Robinson, E. S., Onasch, T. B., Worsnop, D., and Donahue, N. M.: Collection efficiency of α-pinene secondary organic aerosol particles explored via light-scattering single-particle aerosol mass spectrometry, Atmos Meas Tech, 10, 1139−1154, 2017.

Takegawa, N., Miyazaki, Y., Kondo, Y., Komazaki, Y., Miyakawa, T., Jimenez, J. L., Jayne, J. T., Worsnop, D. R., Allan, J. D., and Weber, R. J.: Characterization of an Aerodyne Aerosol Mass Spectrometer (AMS): Intercomparison with other aerosol instruments, Aerosol Sci Tech, 39, 760−770, 2005.

Vaden, T. D., Imre, D., Beránek, J., Shrivastava, M., and Zelenyuk, A.: Evaporation kinetics and phase of laboratory and ambient secondary organic aerosol, P Natl Acad Sci USA, 108, 2190−2195, 2011.

Wagner, R., Höhler, K., Huang, W., Kiselev, A., Möhler, O., Mohr, C., Pajunoja, A., Saathoff, H., Schiebel, T., Shen, X. L., and Virtanen, A.: Heterogeneous ice nucleation of α-pinene SOA particles before and after ice cloud processing, J Geophys Res-Atmos, 122, 4924−4943, 2017.

Yli-Juuti, T., Pajunoja, A., Tikkanen, O. P., Buchholz, A., Faiola, C., Väisänen, O., Hao, L. Q., Kari, E., Peräkylä, O., Garmash, O., Shiraiwa, M., Ehn, M., Lehtinen, K., and Virtanen, A.: Factors controlling the evaporation of secondary organic aerosol from α-pinene ozonolysis, Geophys Res Lett, 44, 2562−2570, 2017.